# A Review on Bitumen Rejuvenation: Mechanisms, Materials, Methods and Perspectives

**Valeria Loise [1] , Paolino Caputo [1],* , Michele Porto [1], Pietro Calandra [2],* ,
Ruggero Angelico [3] and Cesare Oliviero Rossi [1]**

1   Department of Chemistry and Chemical Technologies, University of Calabria, Via P. Bucci, Cubo 14/D,
    87036 Rende (CS), Italy; valeria.loise@unical.it (V.L.); michele.porto@unical.it (M.P.);
    cesare.oliviero@unical.it (C.O.R.)
2   CNR-ISMN, National Council of Research, Institute for the Study of Nanostructured Materials,
    Via Salaria km 29.300, 00015 Monterotondo Stazione (RM), Italy
3   Department of Agricultural, Environmental and Food Sciences (DAAA), University of Molise,
    Via De Sanctis, 86100 Campobasso (CB), Italy; angelico@unimol.it
*   Correspondence: paolino.caputo@unical.it (P.C.); pietro.calandra@ismn.cnr.it (P.C.);
    Tel.: +39-0984-493381 (P.C.); +39-0690-672409 (P.C.)

**Abstract:** This review aims to explore the state of the knowledge and the state-of-the-art regarding bitumen rejuvenation. In particular, attention was paid to clear things up about the rejuvenator mechanism of action. Frequently, the terms rejuvenator and flux oil, or oil (i.e., softening agent) are used as if they were synonymous. According to our knowledge, these two terms refer to substances producing different modifications to the aged bitumen: they can decrease the viscosity (softening agents), or, in addition to this, restore the original microstructure (real rejuvenators). In order to deal with the argument in its entirety, the bitumen is investigated in terms of chemical structure and microstructural features. Proper investigating tools are, therefore, needed to distinguish the different mechanisms of action of the various types of bitumen, so attention is focused on recent research and the use of different investigation techniques to distinguish between various additives. Methods based on organic synthesis can also be used to prepare ad-hoc rejuvenating molecules with higher performances. The interplay of chemical interaction, structural changes and overall effect of the additive is then presented in terms of the modern concepts of complex systems, which furnishes valid arguments to suggest X-ray scattering and Nuclear Magnetic Resonance relaxometry experiments as vanguard and forefront tools to study bitumen. Far from being a standard review, this work represents a critical analysis of the state-of-the-art taking into account for the molecular basis at the origin of the observed behavior. Furnishing a novel viewpoint for the study of bitumen based on the concepts of the complex systems in physics, it constitutes a novel approach for the study of these systems.

**Keywords:** Bitumen; rejuvenator; oils; flux agents; physical chemistry techniques; structure; RAP

---

## 1. Introduction to Bitumen and Ageing

Asphalts are well-known materials used for road pavement throughout the world. They are heterogeneous systems where one phase is constituted by macro-meter sized inorganic particles called aggregates, and the other one is the binding agent (bitumen). The latter, in turn, is a micro heterogeneous complex viscous fluid constituted by nano-meter sized aggregates of polar molecules (asphaltenes) hierarchically organized with different levels of aggregations [1], and dispersed in a more apolar continuous phase of saturated paraffins, aromatic oils and resins called maltene [2,3].

Asphalts are, therefore, biphasic systems, with the predominant phase (93–96%, *w/w*) made by the macro-meter sized inorganic aggregates (size from microns to millimeters) hold together by small amounts of binding bitumen which constitutes the second phase. Although the bitumen constitutes the minor part of the asphalt, it however plays the most important role, giving consistency to the overall material, which is necessary for practical purposes: even slight changes in the bitumen will affect the overall properties of asphalt. Even the usability of asphalt can be traced back to the properties of its bitumen. This work will focus the attention on bitumen as a key ingredient in asphalt. Where necessary, however, some extensions will cover asphalt and related aspects. The aggression of chemicals normally presents in the environment, or ageing can oxidize some of the organic components in the bitumen so that the increase in polar functional groups can cause immobilization of an excessive number of macromolecules and ultimately bitumen embrittlement and asphalt cracks. Other processes causing ageing are loss of volatiles and changes in a molecular organization driven by the spontaneous tendency to reach a stable (equilibrium) thermodynamic state, which obviously depends on the conditions [4]. As a consequence of the mentioned processes taking place in bitumen, the final asphalt is susceptible to fracturing or cracking in thermal or mechanical stresses. Ageing of bitumen, therefore, constitutes a serious problem. Aged asphalt can be re-used in mixtures with new binders, answering the economic need for low-cost production and fully facing obvious environmental concerns. Therefore, recent research has been focused on reclaimed asphalt pavements (RAPs), which really are environmentally friendly alternatives and constitute an economically viable way to afford the costs of binder and aggregates. However, their use is still limited (less than 20% in the new mixtures), due to their low rheological performances (high stiffness and low stress relaxation ability [5] which can cause unexpected premature failure [6]. To overcome these problems, opportune actions must be taken to improve the mechanical and chemical properties of aged bitumen or RAP/bitumen mixtures. The addition of opportune additives is certainly one of the most effective. For example, a compound able to tune the red-ox state of the polar molecules contained in the bitumen can avoid the degrading process. The restoring of a favorable asphaltene/maltene relative ratio by providing more maltene is another solution, since the viscosity of bitumen is related to the fraction of asphaltene [7]. Another strategy can be directed to the stabilization of the supramolecular aggregates mainly made by asphaltene formed in the maltene phase [8,9] acting on their interfacial tension and consequently better dispersing them. In all these cases, a rejuvenator is usually dealt with. With this work, the recent studies carried out in this field will be highlighted in order to shed light on the possible mechanisms of actions of a rejuvenator and to furnish a panoramic view considering both theoretical considerations and applicative aspects. This works will show the state-of-the-art in the use of rejuvenators in bitumen, taking care to highlight also some applications to the rejuvenation of recycled asphalt for a more complete view of the problematics. Differently from standard review papers, this work, prior to presenting all the state-of-the-art works dealing with rejuvenators (Section 4), it proposes a clear, new and marked distinction between different "rejuvenators" according to their effect and influence at the microscopic length-scale individuating techniques and methods of analysis for their distinction (Section 3 "mechanisms of ageing and rejuvenating" and sub-paragraphs). Then, the work will show methods for the synthesis of ever more performing rejuvenators (Section 5) and suggest vanguard techniques of investigation (Section 6).

## 2. Methods for Bitumen Characterizations

The characteristics of bitumen are not trivial at all: it is an organic high-viscosity viscoelastic binding agent which is itself a composite system constituted by nano-meter sized aggregates of polar molecules (asphaltenes) dispersed in a more apolar continuous phase of saturated paraffins, aromatic oils and resins called maltene [2,3]. Usually, these fractions can be determined by the so-called S.A.R.A. determination (Saturates, Aromatics, Resins and Asphaltenes) [10]. In this method, the sample of bitumen is dissolved in peroxide-free tetrahydrofuran solvent (usually to reach a 2% (*w/v*) solution). Saturated components of the sample are developed in *n*-heptane solvent while the aromatics in a 4:1

mixture of toluene and *n*-heptane. Afterwards, the rods are dipped into a third tank, (usually 95 to 5% mixture of dichloromethane and methanol). This organic medium proved suitable to develop the resin fraction, whereas, the asphaltene fraction is left on the lower end of the rods. However, it must also be pointed out that asphaltenes are not classified using their molecular structure, but they are defined traditionally on the basis of the procedure required to extract them from heavy oils [11]. Bitumen are a not well-defined mixture of constituents so different methods of analysis exist where solvents are added to bitumen to determine its chemical properties. For example, Zenke [12] offered a method for not only determining the number of maltenes and asphaltenes of bitumen, but also distinguishing light-, middle-, and high-solubility of asphaltenes. However, for an effective study, the general viewpoint of their overall assembly becomes more important than the detailed chemical speciation of the various molecules involved. Based on this, a testing method for determination of the quality of bitumen, called "simplified laboratory method", was developed in order to gain fast and easy information. S.A.R.A. determination and the attempts by Zenke are examples of this approach to gain chemical information. Bitumen are also regarded in terms of their performances, for which a rheological description of the materials is often given. Empirical approaches are always followed to determine the performances within a chosen temperature range [13,14] for convenient use [15,16]. Due to the general description of the bitumen performances in rheological terms (penetration index, softening point, ductility, viscosity), and due to the need of quick, easy and low-cost methods to characterize the bitumen, the lack of detailed knowledge of the supramolecular assembly characterizing the bitumen structure at the various levels of complexity caused the fact that a rational correlation of the bitumen structure with its performances is actually missing. Attempts at more sophisticated investigative tools are present. Scattering experiments, and in particular X-ray scattering ones, would be advisable to probe the structure from the Å to the meso-scale. As a matter of fact, remarkable attempts have been made since the '60s [17], but the bitumen complex organization has hindered the development of such structural study in details. The structural investigation has been, therefore, generally carried out by Atomic Force Microscopy (AFM) [18], by Confocal Laser Scanning Microscopy [19], by Optical Microscopy [20], and Fluorescence Microscopy [21], but all these methods were used to probe the micro-scale (not going deeper to the nano-scale) and the surface. Attempts at gaining information on the nano-scale structure of the bulk have been limited, and the results remained quite hypothetical [22]. Even the "colloidal structure" is just empirically derived by the contents of aromatics, resins, asphaltenes and saturates [23]. This is probably due, in our opinion, to the always-urgent need of improving performances for applicative purposes so that basic research, highlighting the specific intermolecular interactions and the molecular organization at the base of the observed behavior has been sometimes overlooked. Therefore, there is still a lack of information on how many additives affect the supra-molecular structure and the distribution of aggregates within the bituminous colloidal network and how this can reflect on the overall material properties. This makes the relationship between molecular interactions and the final material structure/properties (which is ultimately the final objective of physical chemistry) still quite vague. This problem was already faced recognizing that bitumen are characterized by intermolecular associations at different length-scales: asphaltene molecules are aggregated to form stacks by self-interactions and these aggregates are stabilized by polar resins, due to their amphiphilic chemical nature and the overall structures are then dispersed in a paraffin-like apolar matrix. These characteristics render the material a truly complex system, so an approach based on the complex systems theory is, therefore, necessary foreseeing different levels of complexity, each of them potentially showing emerging properties arising from the opportune organization of the molecules. It has been, in fact, recently highlighted that asphaltenes tend to form stacks of about 18 Å, organized at higher levels of complexity in anisotropic aggregates of about 200 Å × 28 Å, which, again, are assembled to form micrometer-size elongated aggregates characterized by the so-called "bee-structure" [24]. However, asphaltenes are unable to create a continuous network [25]. These recent findings opened the door to a better comprehension of the bitumen structure from X-ray scattering data, making this technique a method of election [26] for the structural investigation of such systems. In this context, it was shown

that the influence of additives is exerted by their preferential localization in the maltene phase or close to the asphaltene clusters, depending on the additive chemical characteristics, thus, finally affecting the overall rheological properties. An additive can play, therefore, important roles: it can exert its effect at the inorganic/organic interface, or, at a lower level of complexity, within the maltene/asphaltene aggregates, whereas, a redox additive works at the chemical state of the single polar molecules, i.e., at an even lower level of complexity. Another mechanism exerted by the additive is the formation of a network inside the maltene giving elasticity to the bitumen [27] or a simple change of physical phase transition with fluxing at a higher temperature while conferring rigidity at lower ones [28]. The so-called "antistripping agents" act by stabilization of eventual supramolecular asphaltene aggregates in the maltene phase [8,9]. As regards the additive chemical nature, selected polymers have been used (low-density polyethylene, ethylene-vinyl-acetate, SBS-polyphosphoric acid (PPA), Elvaloy, etc.) [29], as well as smaller molecules falling in the categories of organosilane and phospholipids [30] or paraffinic synthetic waxes, derivate of fatty amines and surfactants [31], antioxidants [32], etc. Once introduced the chemical and the structural characteristics of a bitumen, the next paragraphs will highlight the modifications taking place during ageing and how the original state can be restored by the introduction of the concept of rejuvenator.

## 3. Mechanisms of Ageing and Rejuvenating

### 3.1. Ageing

Due to the ageing process of bitumen and its corresponding increase in viscosity, the stiffness of asphalt pavement increases during its lifetime. The general, zero-order, description of aging is given in terms of an overall mechanism where some of the maltene medium is transformed into the asphaltene phase, resulting in higher asphaltene and lower maltene contents. This leads to a higher viscosity and lower ductility, due to the stronger polar-polar interactions between asphaltenes. In simpler words, "when the asphaltene micelles are not sufficiently mobile to flow past one another under the applied stress, the resistance of asphalt binder to cracking or fracture is decreased" as beautifully depicted by J. Petersen [4]. However, ageing is a more complex process involving different sub-mechanisms usually taking place in different time-scales. It can already take place during asphalt construction through volatilization of light components in the maltene. Then, long-term aging occurs in the field as a consequence of different processes:

1.  Oxidative, due to changes in composition through a reaction between bitumen constituents and atmospheric oxygen;
2.  Evaporative, due to the evaporation of low-molecular weight components in the maltene. These compounds have higher vapor pressure and are somehow volatile so they can escape the maltene phase causing not only a change of its composition, but also an overall reduction of its amount in the bitumen;
3.  Structural ageing, by a chemical reaction between molecular components causing polymerization with consequent formation of a structure within the bitumen (thixotropy) [33].

The processes involved are schematically depicted in Figure 1.

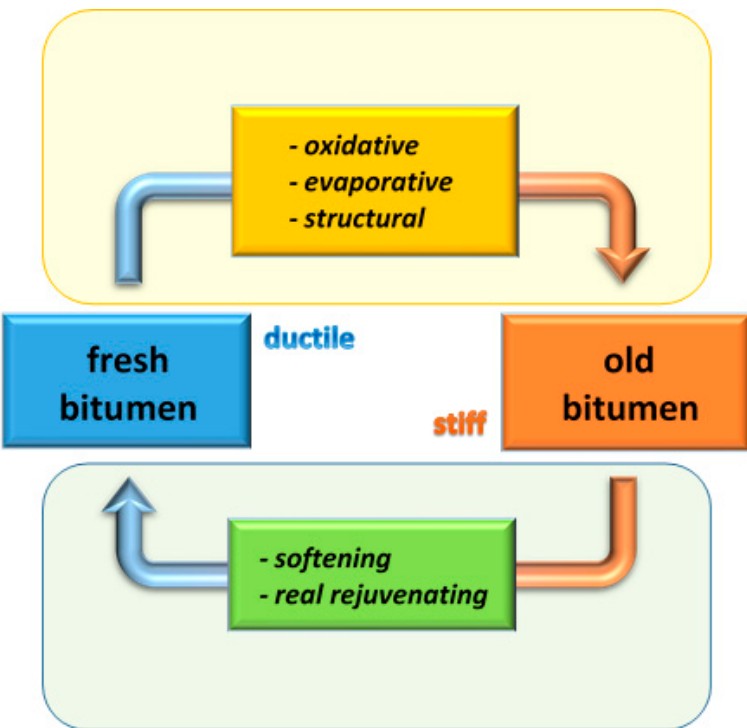

**Figure 1.** Scheme showing the main processes involved in bitumen ageing and its rejuvenation.

*3.2. Rejuvenation*

A rejuvenator has been defined as that agent capable of restoring the original rheological properties of a bitumen. Therefore, it is assumed that the primary action of a rejuvenator should be to bring the bitumen to lower stiffness and viscosity and higher ductility. Despite the above description of bitumen ageing showing different and somehow interconnected mechanisms, taking place, simultaneously, the very first action to restore the original bitumen rheological properties is to shift back the ratio between solid asphaltenes and fluid maltenes to higher maltene contents. However, a rheological rejuvenating agent can exert its action in different sub-mechanisms (see the schematic diagram of Figure 1):

1.  Softening (usually called fluxing) agent: flux oil, lube stock, slurry oil, etc. can lower the viscosity of the aged binder;
2.  Real rejuvenator: it helps to restore the physical and chemical properties [33].

As it will be seen (see Section 4), literature does not make a clear distinction among the types of mechanism exerted by the specific rejuvenator: this name is usually given to any king of additive, which allows a certain restoring of the original rheological properties, no matter the effective mechanism involved. Instead, we are keen on making a clear distinction: given the aim of this contribution to go deeper into the details and to make clear all the aspects involved, it will be now defined a rheological rejuvenator as any type of additive causing a certain restoring of the rheological characteristics. In the specific, if the rejuvenator is also able to restore even the bitumen inner structure, then it will be defined as a real rejuvenator. If, on the other hand, the rheological rejuvenator makes the bitumen more ductile and less viscous/brittle by simply furnishing oily components to maltene without restoring the original complex structure (hierarchical structures of asphaltenes), then it will be called as softening (more often fluxing) agent. It can be seen that in this distinction, the term "rheological rejuvenator" has a more general sense and is associated with the concept of "rejuvenator" usually present in the literature. For this reason, in the following, when simply naming "rejuvenator" it will be referred to rheological rejuvenator, to resemble the classical meaning of rejuvenator usually adopted. Instead, when dealing with real rejuvenator effect, the terminology "real rejuvenator" will be stressed. Rheological rejuvenators are, however, usually based on oils: lubricating oil extracts and extender

oils. They contain an adequate amount of maltene constituents, naphthenic or polar aromatic fractions, which re-balance the composition of the aged binder in favour of such compounds or those usually lost during construction and service. Rejuvenator must fulfil two requirements:

1.  They should have a high proportion of aromatics, which are necessary to keep the asphaltenes dispersed;
2.  They should contain a low content of saturates, which are highly incompatible with the asphaltenes.

### 3.2.1. Down to the Nano-Scale

Of course, the effectiveness of a rejuvenation action depends on the uniform dispersion of the rejuvenator within the bitumen. In fact, after its mixing with aged bitumen and the obvious homogenization, diffusional processes taking place at longer times and shorter length-scales must complete the job. This aspect was first faced by Lee et al. in 1983 [34] who mixed a dye with the rejuvenator to estimate the homogenization of the overall product by visual inspection. The authors concluded that mechanical mixing could give a uniform distribution of the rejuvenators within the bitumen. However, our perplexity holds in the fact that visual inspection cannot probe the uniformity of dispersion at the sub-micro scale, which instead is the final goal of an efficient mixing allowing complete rejuvenator action. According to a work by Carpenter and Wolosick [35] published few years before, the diffusion of a rejuvenator into an aged binder is a complex, multi-step mechanism consisting of four steps (the curious reader is redirected to the reading of that article for details), but basically the requirement of a complete homogenization of the rejuvenator down to the nano-scale was already pointed out. However, their results were later confirmed by Noureldin and Wood [36], and Huang et al. [37]. The complete homogenization, down to the sub-micro scale cannot happen without being diffusion-driven and diffusion-limited—a dynamical process whose rate is given by the viscosity of the medium. This indeed, has been shown by Karlsson and Isacsson [38], who highlighted that the diffusion rate is governed by the viscosity of the maltene phase rather than the viscosity of the recycled binder as a whole, an aspect which was somehow observed previously by Oliver [39] who had suggested that the diffusion could be sped up by diluent oil fractions addition and/or raising the temperatures.

### 3.2.2. Distinguishing Softening Agents and Real Rejuvenators

From the point of view of the physical performances, softening (fluxing) and real rejuvenating, which constitute two different mechanisms, can be distinguished experimentally. Whereas, the stiffness or rigidity of a bitumen, usually empirically determined by simplified and fast methods generally by immediate techniques developed for characterizing mechanical and rheological properties and used especially in the field of engineering [40], the distinction of the two effects of fluxing and real rejuvenating needs the exploitation of methods with a substantial physical basis. In this ambit, small amplitude oscillatory rheometry is a useful technique using specific specimen geometries and mathematical interpretation of the data to achieve physical quantities: the complex modulus $G^*$. $G^*$ is a measure of the total energy required to deform the specimen and is defined as:

$$|G^*|^2 = (G')^2 + (G'')^2 \tag{1}$$

where $G'$ is the elastic modulus (or storage modulus), a measure of the energy stored in the material during oscillation, and $G''$ is the viscous modulus (or loss modulus), a measure of the energy dissipated as heat. Martin Radenberg et al. [41] highlighted the difference between a "rejuvenating" (here associated with our conception of real rejuvenating) effect and a "fluxing" agent from a rheological perspective using the so-called black diagrams, which depict the magnitude of the complex modulus $G^*$ versus the phase angle ($\delta$, defined as $\tan \delta = G''/G'$) obtained from the dynamic tests. In black diagrams, frequency and temperature are eliminated. This method was previously suggested by Airey and Brown [42] to assess and compare the rheological properties of bitumen. The characterization of

the two different actions of rejuvenator has been recently carried out by NMR. Although bitumen is a complex material, ambitious studies are facing structural characterization by probing relaxation times. Application of Low-Field NMR has been used for the determination of physical properties of petroleum fractions [43,44], and the Inverse Laplace Transform (ILT) analysis of the NMR echo signal decay gives the $T_2$ relaxation time which can be connected to different domains characterized by different rigidities [45]. The chemical reasoning for this lies on the molecular constraint, causing dynamic hindrance and lowering $T_2$, an effect that can be considered quite general and already found also in different systems [46]. These attempts have given interesting and encouraging results, which deserve to be tailored. We support efforts in this direction since the comprehension of the microscopic/molecular processes at the basis of the observed behavior of a material is of fundamental importance for the improvement of materials characteristics in specific applications.

## 4. The State-Of-The-Art

### 4.1. General Requirements

As a general requirement, additives should be non-hazardous and stable over a wide range of temperatures, from production to application. In addition, they must not experience any exudation or evaporation, in order to ensure good performance over the designed lifetime of the asphalt pavement. Further specific requirements depend on the local country specification. Bocci et al. for example [47], focus on the use of a specific bio-based rejuvenator to produce asphalt concrete using a high amount of RAP without scarifying the mix performance and complying with the Italian specifications. In particular, the paper presents the results from a trial section on a highway connecting Ancona to Perugia, in the center of Italy.

In any case the use of "forbidden" chemicals should not be the goal of the work: for example, Tine et al. [48] take care in clarifying that the additives they studied in their work (a liquid process oil with a typical viscosity at 40 °C of 700 mm$^2$/s) conform to all current Registration, Evaluation, Authorisation and restriction of Chemicals, (REACH) and Polycyclic Aromatic Hydrocarbons (PAHs) limits. Another additive they used (a tall oil distillate originating from pine trees) is "not classified as dangerous according to Directive 67/548/EEC and its amendments and according to EU legislation". In our opinion, any work should clarify the environmental/safety/legal issues connected to the additives presented.

In this sense, Król et al. [49], and Somé et al. [50] show that the addition of some particular bio-additives has a large potential application as reversible fluxing agents in bitumen industry, RAP technology and Warm Mix Asphalt (WMA). They used vegetable oil produced using various raw materials (Rapeseed oils, Soybean oil, Sunflower oil, Linseed oil, etc.) and combined in bituminous binders generally as a modifier/additive (up to 10%). Interestingly, not only they used vegetable oils, but they also perform chemical action to prepare other additives (they use methyl esters of fatty acid obtained by transesterification of vegetable oils). The aspect of a couple the use of raw/cheap materials with chemical actions to increase their performances is intriguing and promising. The chemical actions that can be performed for the optimization of additive performances will be discussed in Section 5.

The rejuvenating benefits can be utilized to allow for higher RAP addition using traditional hot mix asphalt (HMA) configuration or in the warm mix asphalt (WMA) technology since the additives, changing the viscosity of the binder, allow for the lowering of the asphalt mixture production temperature. Critical issue related to the application of the additive fluxing of the bitumen is the final viscosity of the bitumen and stiffness of the mixture placed in the pavement. In fact, while fluxing is desired during mixture production, placement and compaction, it is no longer appreciated once the road is open to traffic. Considering the examples of Król et al. [49], and Somé et al. [50], the suitability criterion for vegetable oils and the corresponding methyl esters for obtaining environment-friendly bitumen fluxes is their reactivity to the oxypolymerization reaction, which raises the viscosity of the stock, thus, contributing to its hardening and drying. The efficiency of polymerization depends on the

number of double bonds and their position in the aliphatic chain of fatty acid. Oxypolymerization of the fatty acids present in the vegetable stocks is a multistage process and leads to the crosslinking of their structural units. The principle of using additives susceptible to oxypolimerization has been recently exploited for preparing additives, increasing the bitumen viscosity [51].

This paragraph, from the next sub-paragraph on, is devoted to showing the works involving rejuvenators as additives in bitumen. Some applications to the rejuvenation of partly recycled asphalt will also be shown for a more complete view of the problematics also because for practical uses sometimes rejuvenation must be exerted on recycled asphalt (RAP, reclaimed asphalt pavement). They have been grouped according to the experimental methods that can be used for the rejuvenation evaluation. In addition, particular care has been taken in using the same names and formalisms used by the various Authors, for a better comparison with the original works.

*4.2. Rejuvenation Probed by IR*

Zargar et al. in 2012 [52] explored the possibility of using a waste cooking oil (WCO) as rejuvenator in the aged bitumen, in order to reduce the expense of highway renovation. They added various amounts of a WCO mainly composed by Oleic acid (C18:2n9c, 43.7%), Palmitic acid (C16:0, 38.4%) and Linoleic acid (C18:2n6c, 11.4%) to an aged bitumen monitoring the changes in standard parameters like penetration index, softening point, Brookfield viscosity and dynamic shear viscosity. The comparison among pristine, aged and rejuvenated bitumen gave self-consistent results furnishing clear clues: the aged bitumen, which is stiffer than the virgin one as result of oxidation processes, can be substantially driven to the performances typical of the virgin bitumen by progressive addition of waste cooking oil. The restoring of the virgin performances takes place at WCO content of 3%, but they can even be improved (n penetration index, softening point and viscosity) by further addition of WCO up to 5%. See, for example, Figure 2 where the penetration value versus various rejuvenator amounts is reported. These results were interpreted at the molecular basis with the aid of Fourier transform infrared spectroscopy (FT-IR) which probes the amount of oxidized functional groups, and specifically C=O and S=O. The increase of polar functional group signal intensity, and specifically C=O and S=O ones, in FT-IR spectra when the bitumen is aged correlates with its stiffness increase. This suggests that the loss in bitumen performances is due to the oxidation process. Even, this increase correlates with an increase in asphaltenes fraction, i.e., the most polar component of the bitumen. On the other hand, the addition of WCO implies a substantial restoring of the pristine C=O and S=O signal, together with the pristine asphaltene/maltene ratio, although the process seems not to be complete/quantitative. Typical IR spectra, together with the most important band attributions, are reported in Figure 3. In conclusion, WCO appears to have a rejuvenating effect because it reduces the oxidation (asphaltene content) compared to an aged bitumen, even if it is not capable of restoring the correct ratio maltene asphaltenes. However, it is interesting to point out that the rejuvenated bitumen has less tendency to ageing compared to original bitumen and has lower volatility than the virgin bitumen reasonably, due to the lower volatility of bio-oil as a consequence of its high content in saturated hydrocarbons. Unfortunately, the Authors do not give details on the FT-IR experiments.

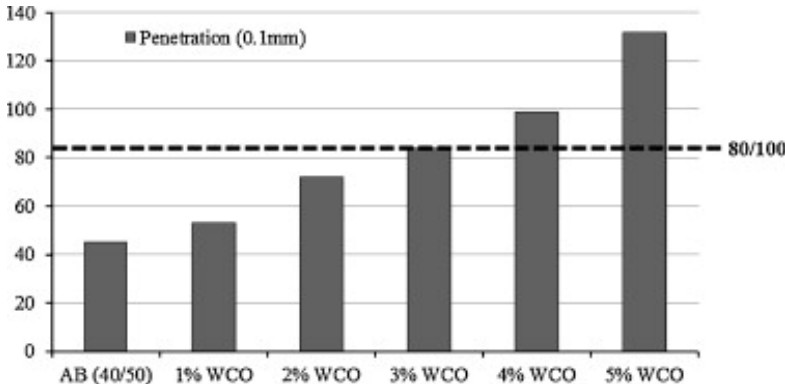

**Figure 2.** Penetration value versus different rejuvenated waste cooking oil (WCO) bitumen's (AB, aged bitumen) (reprinted from Reference [52] with the permission of Elsevier).

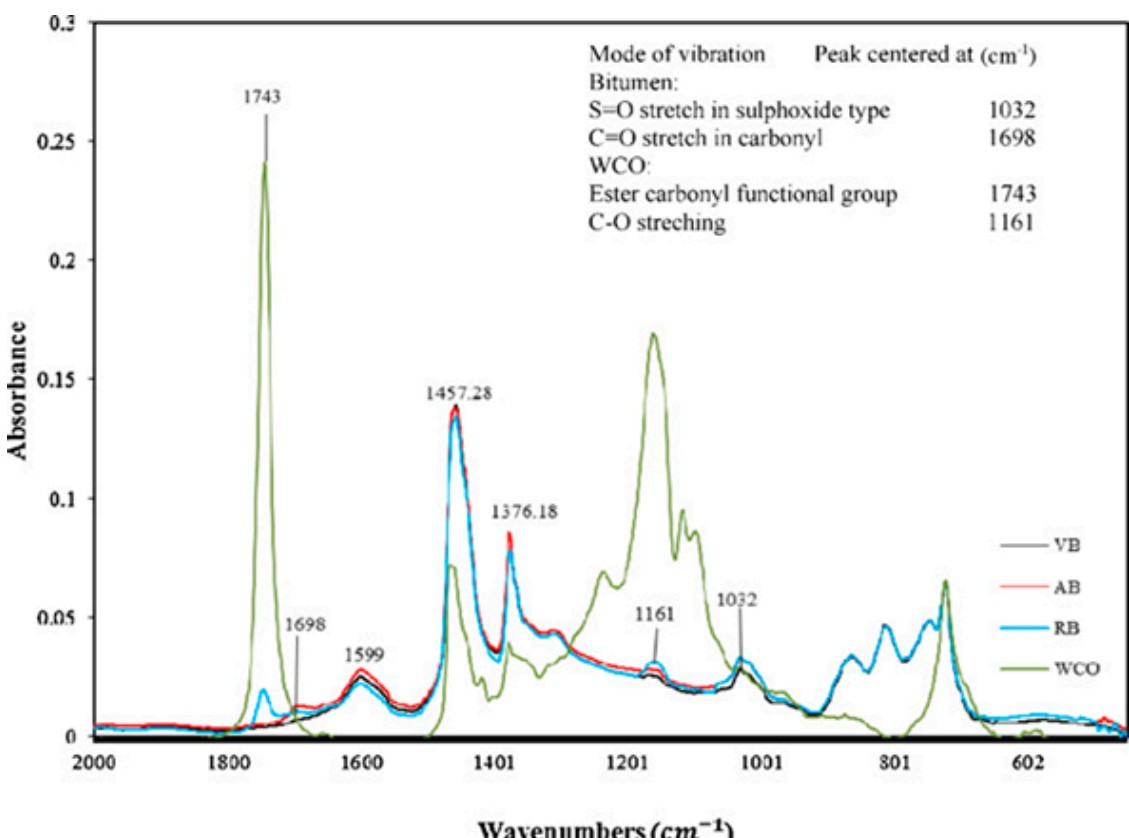

**Figure 3.** FTIR analysis of virgin aged and rejuvenated bitumen (VB, virgin bitumen; AB, aged bitumen; RB, rejuvenated bitumen) (reprinted from Reference [52] with the permission of Elsevier).

To avoid FT-IR artefacts, due to differences in the optical path (different thickness of samples) or in experimental procedures, Nayak and Sahoo [53] wisely consider the relative amount of ketonic bond with respect to the signals of functional groups which are not expected to change, like the bending of C–H bonds. This peak is pointed out in Figure 4.

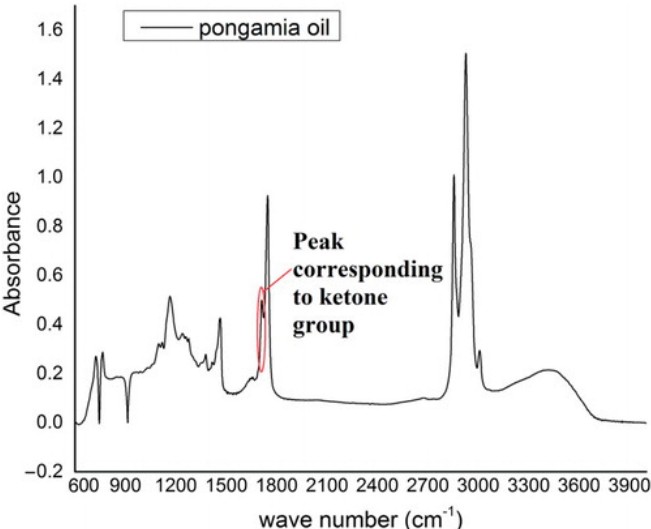

**Figure 4.** FTIR spectrum for pongamia oil (reprinted from Reference [53] with the permission of Taylor and Francis).

This criterion of using a relative intensity was proposed by De la Roche et al. [54], but the use of the relative area is consistent with the consideration by Lamontagne et al. [55].

In any case, Nayak and Sahoo studied the rejuvenating effect of two different types of oils (the pongamia oil and a composite castor oil made of 70% castor oil and 30% coke-oven gas condensate). The results confirm the clues by Zargar et al. showing an increase of the C=O relative signal when the bitumen is aged. Moreover, they found that the C=O relative signal slightly tends to decrease when the composite oil is added, but more markedly increases when pongamia oil is added. The reason lies in the fact that the ketone group is present in pongamia oil itself, and therefore, it is not necessarily due to ageing process. This result should be interpreted as an invitation in being prudent when performing FT-IR experiments and when interpreting its results.

### 4.3. Thermal Stability Helps in Probing Rejuvenation

By monitoring viscoelasticity, creep and response, FT-IR, Thermogravimetric Analysis (TGA) and Differential Scanning Calorimetry (DSC), Nayak and Sahoo proved that successful rejuvenation takes place when these oils are added to the bitumen. It is interesting to notice that both oils have good thermostability performances. The virgin binder showed a visible mass loss at temperatures below 200 °C; instead, all the rejuvenated binders along with aged binder showed significant mass loss beyond 200 °C, but such temperatures are much higher than those typically adopted in hot mix asphalt preparation, rendering this difference not significant for practical purposes.

Also, Elkashef et al. [56] used thermal stability analysis to evaluate the effect of rejuvenating soybean oil on reclaimed asphalt pavement (RAP). Acknowledging the importance of studying thermal stability of rejuvenated binders to assure that eventual loss of rejuvenating additive (typically characterized by low molecular mass) does not modify asphalt performance, they carried out thermogravimetric analysis on the RAP, on the blend made by RAP and the pristine bitumen (PG58-28), and finally, on the mixture formed by RAP, soybean oil (12%) and the pristine bitumen. Coupled to Thermogravimetry (TG), an FT-IR spectrometric investigation on the gases produced by heating was also carried out. The Authors' results obtained by Thermogravimetry (TG) can be summarized in Table 1.

**Table 1.** TG results of the studied asphalt binders (reprinted from Reference [56] with the permission of Springer Nature).

| Sample | IDT (°C) | Char Yield (%) | Residue (%) |
|---|---|---|---|
| RAP | 316 | 30 | 7 |
| RAP + PG58-28 | 309 | 26 | 6 |
| RAP + 12% Mod PG58-28 | 309 | 26 | 6 |

This table shows the Initial Decomposition Temperature (IDT), defined as the temperature where 2% mass loss occurs; the char yield percentage is the mass remaining at the temperature of 550 °C, and the percentage residue is the remaining percentage of asphalt material at the end of the analysis. From this table it is clear that the RAP binder shows the highest thermal stability, confirming that the stiffness is due to asphaltene amount that is more thermally stable, whereas, the other two binders show the same behavior, having an IDT few degrees lower. Elkashef et al. also compared the spectra (at a temperature of 390 °C) of the gases evolved from the three different binders, taking into account for the blank constituted by the bare soybean oil. It turned out that the spectrum of the binder containing the rejuvenator shows a relatively intense peak typical of the rejuvenator itself. Although this observation can be thought as trivial, a pseudo-quantitative analysis, carried out at different temperatures by normalizing the FT-IR spectra at the same C-H stretching peak height (2930 $cm^{-1}$), allowed the Authors to conclude that FT-IR spectrometry can be used as a tool to probe the rate of mass loss of the rejuvenator. In fact, a higher mass loss triggered by a temperature increase implies higher relative intensities of the characteristic functional groups of the rejuvenator itself (C=O stretching at 1736 $cm^{-1}$ and the two C-O stretching at 1015 and 1153 $cm^{-1}$).

### 4.4. Rejuvenation May Be Uncorrelated with IR Functional Groups: Need of Chromatography

Cavalli et al. [57] found an interesting aspect. In fact, these Authors pointed out that, despite the addition of rejuvenators, the bitumen physico-chemical oxidation did not reverse: mechanical changes were not caused by chemical changes at functional groups level, but by a rearrangement of polar/nonpolar components. They took into account for three different commercial bio-based rejuvenators: a natural seed oil (called rejuvenator "A"), a cashew nut shell oil (called rejuvenator "B") and a rejuvenator based on tall oil (rejuvenator "C"). After highlighting by FT-IR that rejuvenator A and C have a rather similar chemical nature (see Figure 5 which reports the FT-IR spectra of the bare three rejuvenators), they performed FT-IR spectra on samples of RAP containing 5% of each rejuvenating and compared them to the spectrum of a virgin bitumen 50/70. The results are shown in Figure 6: the peaks corresponding to the ageing of binders, due to oxidation do not disappear although rejuvenators were added to the RAP binder. Chemical structures of RAP +5% C and RAP +5% A were found to be similar, suggesting that the chemical effect of rejuvenator C and A on RAP binder is similar.

The Authors also determined the chemical ageing index (CAI). This is given by the sum of carbonyl index (CI) and sulfoxide index (SI) defined, as suggested in Marsac et al. [58], as the area of the carbonyl and sulphoxide signal, respectively, normalized, to the peaks related to the asymmetric vibration of $CH_2$ and $CH_3$ (around 1455 $cm^{-1}$) and to the symmetric deformation vibration of $CH_3$ (around 1376 $cm^{-1}$) as the latter areas do not change significantly. It must be noted that here the Authors use the same indices as Nayak and Sahoo, by considering the area of the FT-IR signals (and not the bare intensities) and normalizing them to signals related to the C-H functional group since they are expected to be hardly sensitive to the chemical environment.

By using these indices, the Authors found that the unaged binders RAP + 5% A and RAP + 5% C have a higher CAI index than the plain RAP. Most probably due to the fact that seed oil and tall oil themselves contain carboxylic groups C=O, they found the same clues derived by Nayak and Sahoo who found that the ketone group is present in their rejuvenator (pongamia oil) itself.

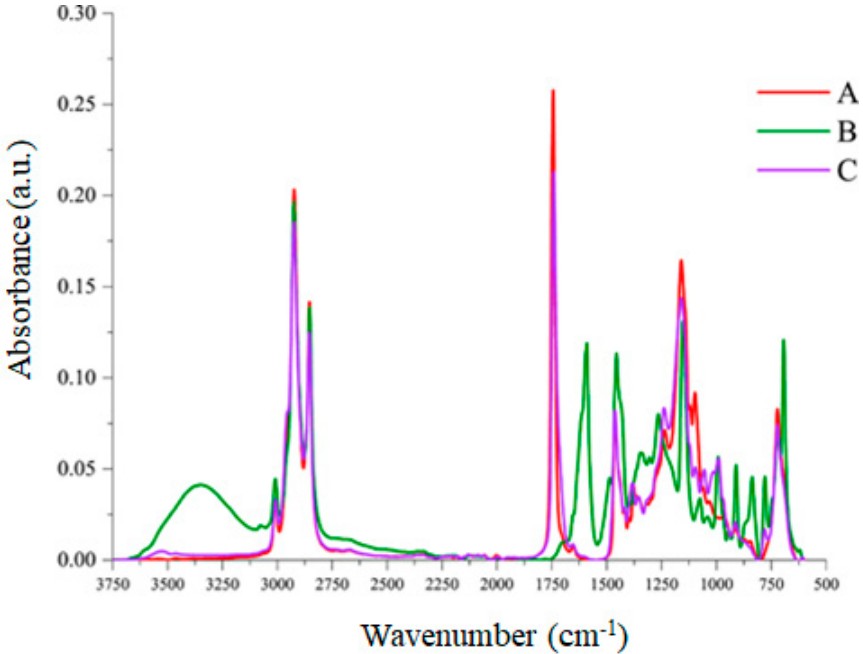

**Figure 5.** ATR-FT-IR spectra of the plain rejuvenators A, B and C before ageing) (reprinted from Reference [57] with the permission of Elsevier).

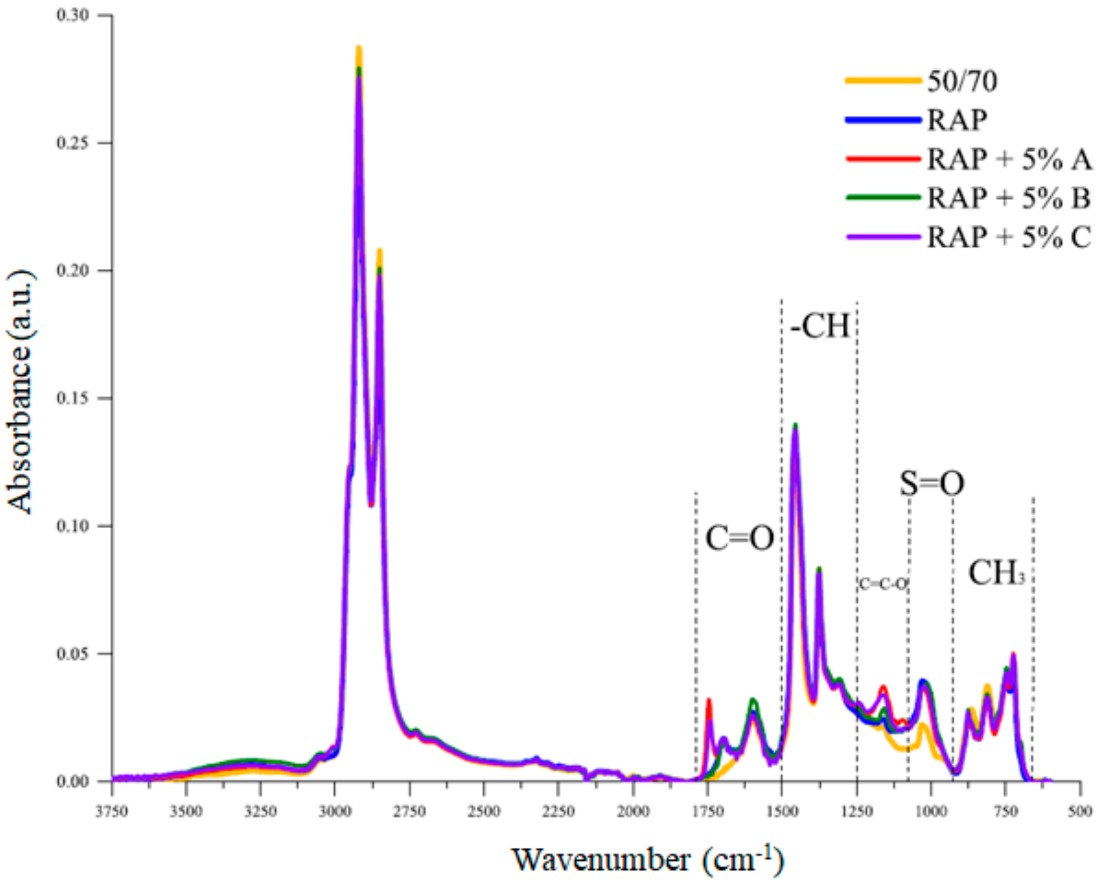

**Figure 6.** ATR-FTIR spectra of the virgin binder 50/70, the RAP binder and the RAP binders with rejuvenators A, B and C (5% by mass of RAP binder each) (reprinted from Reference [57] with the permission of Elsevier).

Elkashef et al. [59] also used the FT-IR-ATR analysis to explore the effect of soybean-derived rejuvenator on two different kinds of performance grade bitumen. The types of bitumen, PG 64-28 and PG 58-28, were aged by Rolling Thin Film Oven Test (RTFOT, similar but slightly different in parameters values from TFOT technique) and pressure-aging-vessel (PAV) methods, and then they were doped with 0.75% of soybean-derived rejuvenator.

Figure 7a shows the increase in the carbonyl index with ageing. The soybean additive does not influence the ageing behavior of the binder. Regarding the sulfoxide index, (shown in Figure 7b) it increases with the aging of PG 58-28. On the contrary, it decreases dramatically to the last step of PG 64-28 ageing (PAV). The drop is also present for the aged doped binder. Ageing was probed by FT-IR-ATR in both the control and modified asphalt binders. As a result, the carbonyl and sulfoxide functional groups increase with ageing. The time evolution of these two functional groups, leads to the conclusion that the modification does not cause any significant influence on the ageing behavior of the asphalt binders. Cavalli et al. also investigated the molecular size distribution with Gel Permeation Chromatography (GPC) in order to evaluate if the rejuvenator modified molecular properties of RAP binder. They used two kinds of detector: one is the refractive index (RI) detector, and the second one is a variable wavelength detector with UV signal (UV). From RI detector, RAP + 5% A and RAP + 5% C (see Figure 8) displayed a shift from the large to the middle size as compared to RAP binder.

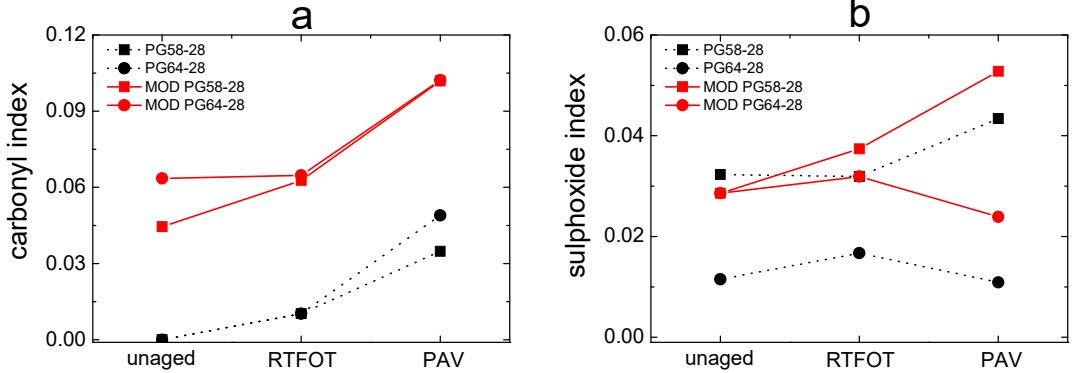

**Figure 7.** Carbonyl index (**a**) and sulfoxide index (**b**) (data taken from Reference [59]).

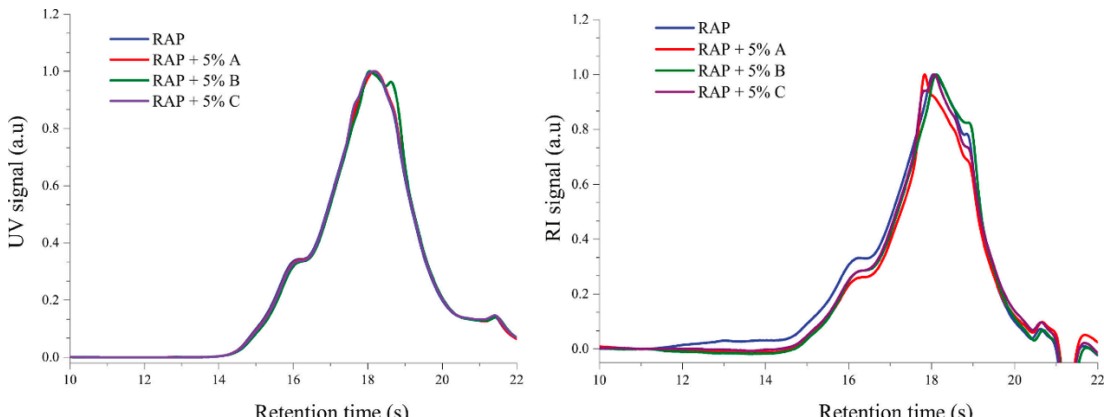

**Figure 8.** GPC spectra with UV (**left**) and RI-signals (**right**) of the RAP binder and the RAP binder with rejuvenator A, B and C (5% by mass of RAP binder each) (reprinted from Reference [57] with the permission of Elsevier).

Comparing the FT-IR spectra and the results obtained with GPC they hypothesized the presence of an ether group (C-O-C) in the RAP modified with A and C, whereas, from UV detector a shift towards the smaller molecular sizes has been observed in RAP + 5% B. In summary rejuvenators could partially change the molecular size distribution of the RAP binders.

The idea of combining FT-IR to obtain information on functional groups and GPC test to obtain information on molecular weight distribution was also exploited by Zhu et al. [60] who studied the effect of a bio-rejuvenator made by a by-product in cotton-oil production and dibutyl phthalate (7.5 wt% in the bio-rejuvenator) as plasticizer. In fact, they carried out a comparison between the FT-IR spectra of pure bio-rejuvenator; pristine bitumen; bitumen aged by Thin Film Oven Test (TFOT - 5 h, 163 °C) followed by a pressure-aging-vessel (PAV) test (aging temperature in the PAV test 90 °C, pressure 2.1 MPa, duration of the test 20 h); PAV was added with 5% or 10% bio-rejuvenator. Moreover, they carried out FT-IR spectra on an asphalt modified with styrene-butadiene-styrene (SBS); SBS-modified PAV, and 5% or 10% bio-rejuvenator on SBS-modified PAV. The peak-area intensity of the oxygenated groups (CO and SO) was used to probe the degree of aging and rejuvenation of the asphalt. Here, again, the Authors calculated the $I_{co}$ and the $I_{so}$ indexes. As expected, for pristine bitumen and SBS-modified asphalt, the carbonyl and sulfoxide indexes increase after aging. As it can be seen by a perusal of Figure 9, both indexes decrease with the addition of the bio-rejuvenator.

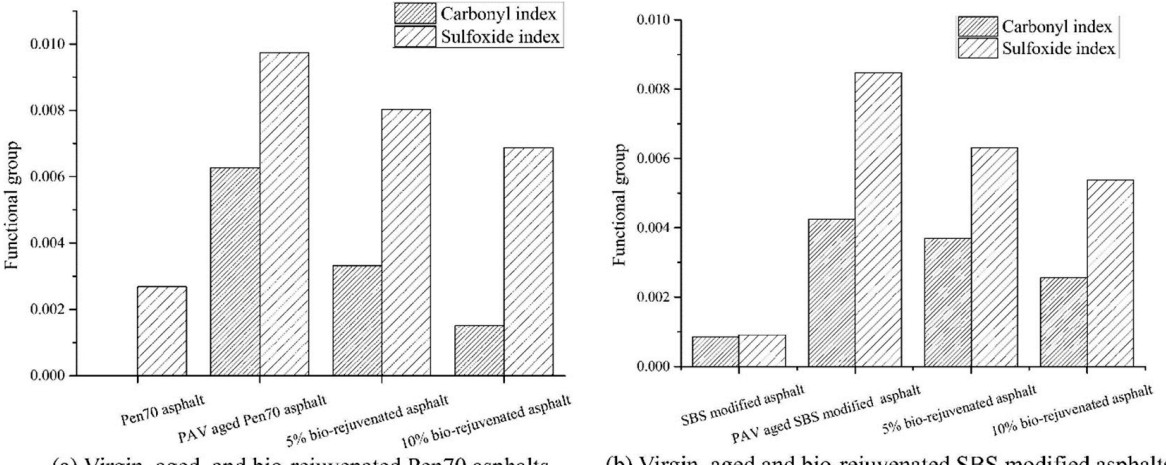

(a) Virgin, aged, and bio-rejuvenated Pen70 asphalts    (b) Virgin, aged and bio-rejuvenated SBS modified asphalts

**Figure 9.** Carbonyl and sulfoxide indexes of virgin, aged, and bio-rejuvenated asphalt (reprinted from Reference [60] with the permission of Elsevier).

However, the carbonyl and sulfoxide indexes of the PAV-aged asphalt cannot be restored to their original levels. The Authors applied the GPC to the system under consideration to shed light on the effect of the rejuvenator on the molecular size distribution. The chromatogram reported in Figure 10a shows that the dw/dlogM versus Mw (weight-average molecular weight) curves for the bio-rejuvenator. Mw of the bio-rejuvenator is about 1000 g/mol, and the narrow and sharp peak indicates that the molecular-distribution dispersity of the bio-rejuvenator is low.

Figure 10b shows that the PAV-aging process causes a decrease in the low and medium-weight molecular content of the pristine bitumen while increasing its high-weight molecular content. Adding the rejuvenator helps in improving the properties of the PAV-aged bitumen by increasing the medium-weight molecular content and decreasing the low-weight molecular content. Figure 10c shows a similar phenomenon with the single-peak in the PAV-aged SBS-modified asphalt splitting into two peaks after bio-rejuvenation. In order to characterize the molecular mass distribution, the Authors used analysis of polydispersity, defined as the Mw-to-Mn ratio (Mw, weight average molecular weight; Mn, number-average molecular weight). The results are reported in Figure 11. Both Mw and polydispersity of the virgin asphalt were found to increase after PAV ageing, due to the various chemical reactions in the asphalt/polymer. On the other hand, the values of the PAV-aged virgin and SBS-modified bitumen show that the rejuvenation effect can be primarily attributed to the changes in the molecular polydispersity of the asphalt.

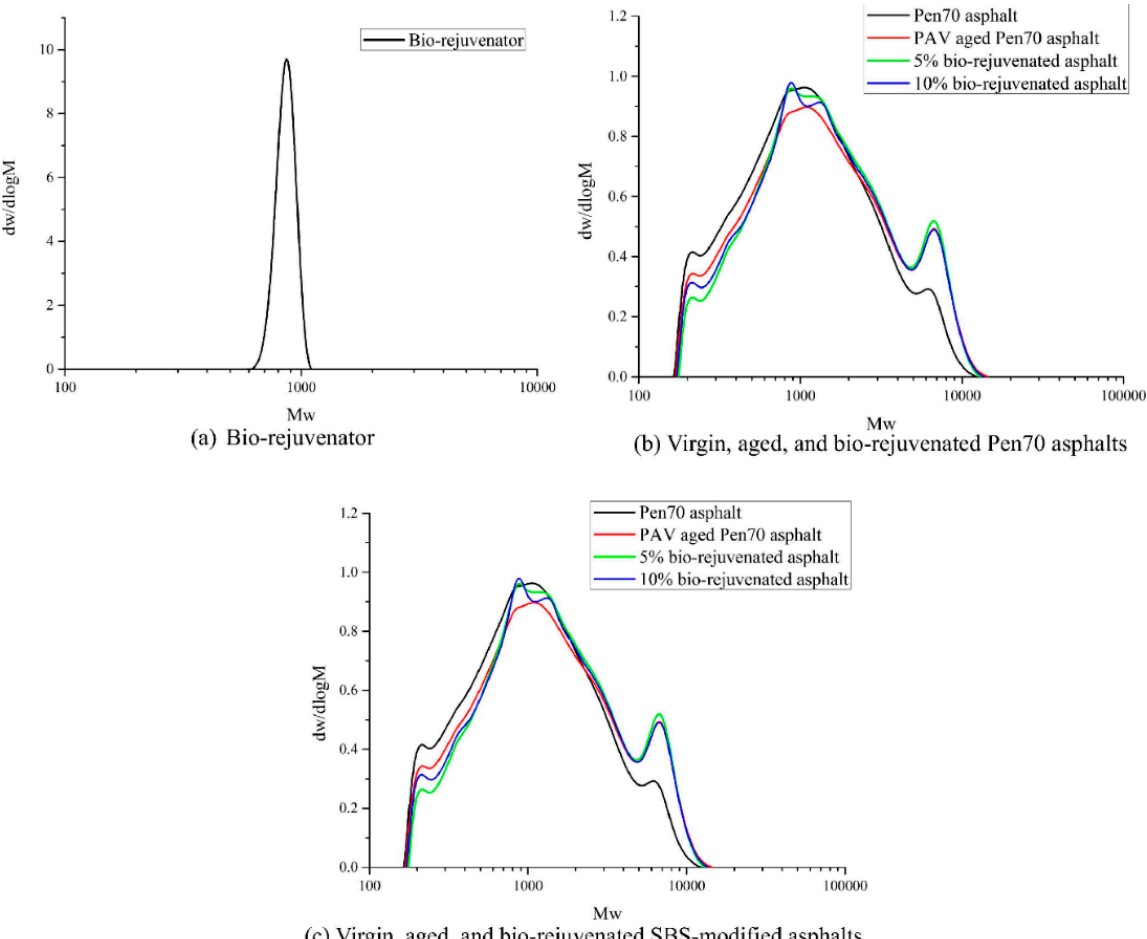

**Figure 10.** GPC curves of bio-rejuvenator and virgin, aged, and bio-rejuvenated asphalt (reprinted from Reference [60] with the permission of Elsevier).

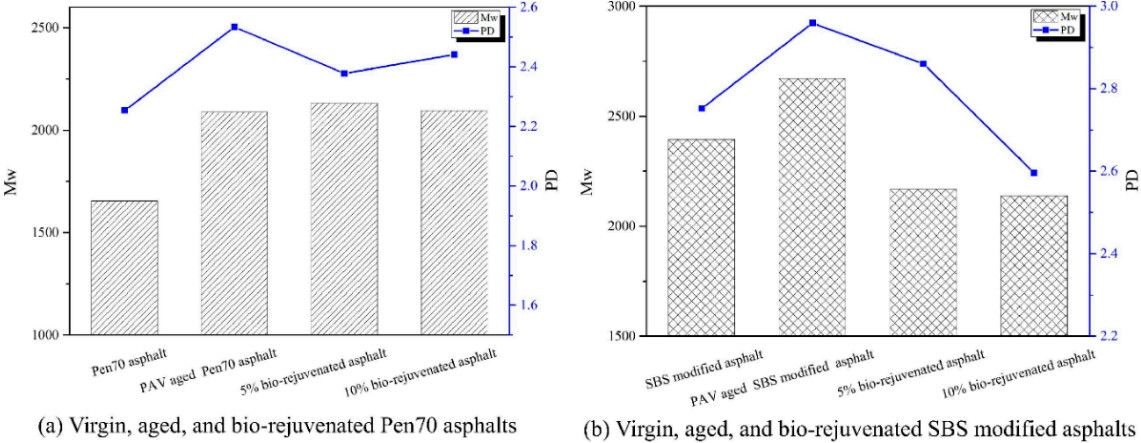

**Figure 11.** GPC Mw and PD results of virgin, aged, and bio-rejuvenated asphalt (for bio-rejuvenator: Mw = 861, PD = 1.0082) (reprinted from Reference [60] with the permission of Elsevier).

Conversely, for the PAV-aged SBS-modified asphalt, Mw decreases with the increase in the dosage of the bio-rejuvenator from 0 to 10%. Both the 5% and 10%-bio-rejuvenated asphalt exhibit lower PDs than the PAV-aged SBS-modified asphalt. However, there is no clear difference between the Mw values of the 5% and 10%-bio-rejuvenated asphalt.

In order to understand the role played by the rejuvenator on the RAP bitumen, Elkashef et al. [61] used the GC-MS (Gas Chromatography-Mass Spectroscopy). They considered a rejuvenator produced from soybean oil, but they did not furnish further details. First, they analyzed the pure rejuvenator: its total ion chromatogram (see Figure 12a) shows five distinct and well-resolved peaks. The pure rejuvenator was then subjected to RTFOT-aging and PAV-aging following the same protocol as that used for asphalt binders aging to assess the chemical stability of the rejuvenator with aging. The total ion chromatogram for the RTFOT-aged and PAV-aged rejuvenator samples are shown in Figure 12 (panels b and c, respectively): they clearly show that the aged rejuvenator gives the same peaks at the same retention times and with similar relative intensity as the unaged rejuvenator. This indicates that the chemical composition of the rejuvenator is preserved during aging.

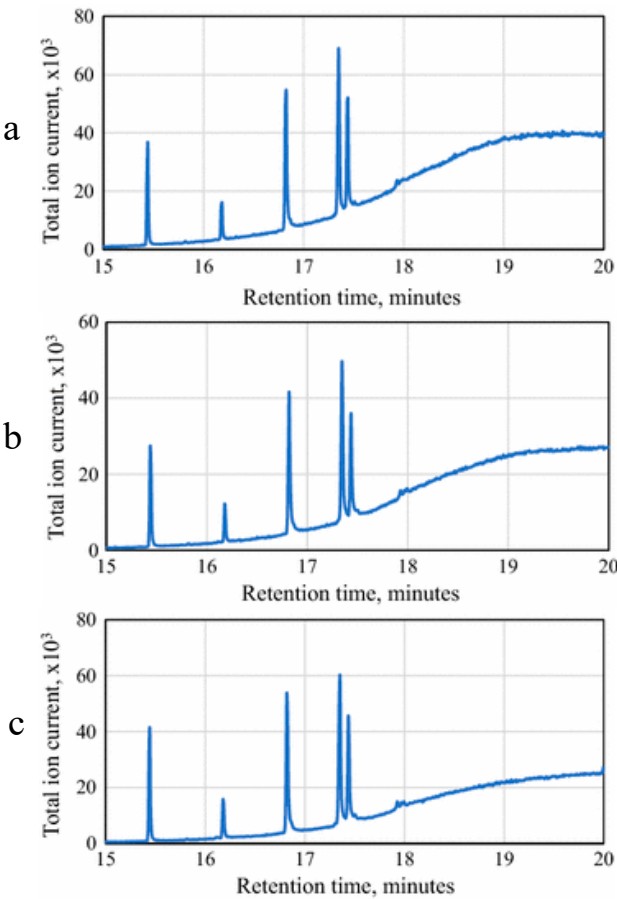

**Figure 12.** Total ion chromatogram for the used rejuvenator (**a**), the RTFOT-aged rejuvenator (**b**) and the pressure-aging-vessel (PAV)-aged rejuvenator (**c**) (reprinted from Reference [61] with the permission of Elsevier).

Subsequently, they submitted rejuvenated RAP (unaged, RTFOT-aged and PAV-aged) to pyrolysis to analyze the evolved gases using Gas Chromatography coupled with Mass Spectrometry (GC-MS) (see Figure 13). The total ion chromatogram of the unaged rejuvenated binder shows the rejuvenator's peaks in addition to other smaller peaks attributed to the binder itself. However, the structure of the rejuvenator added to the binder appears to change with aging.

It is, therefore, clear that the rejuvenator interacts with the RAP binder, with the consequent structure modification of the rejuvenator itself. Strikingly, the two peaks at 17.3 and 17.4 min are reduced in intensity with aging, and they entirely disappear in the PAV-aged binder.

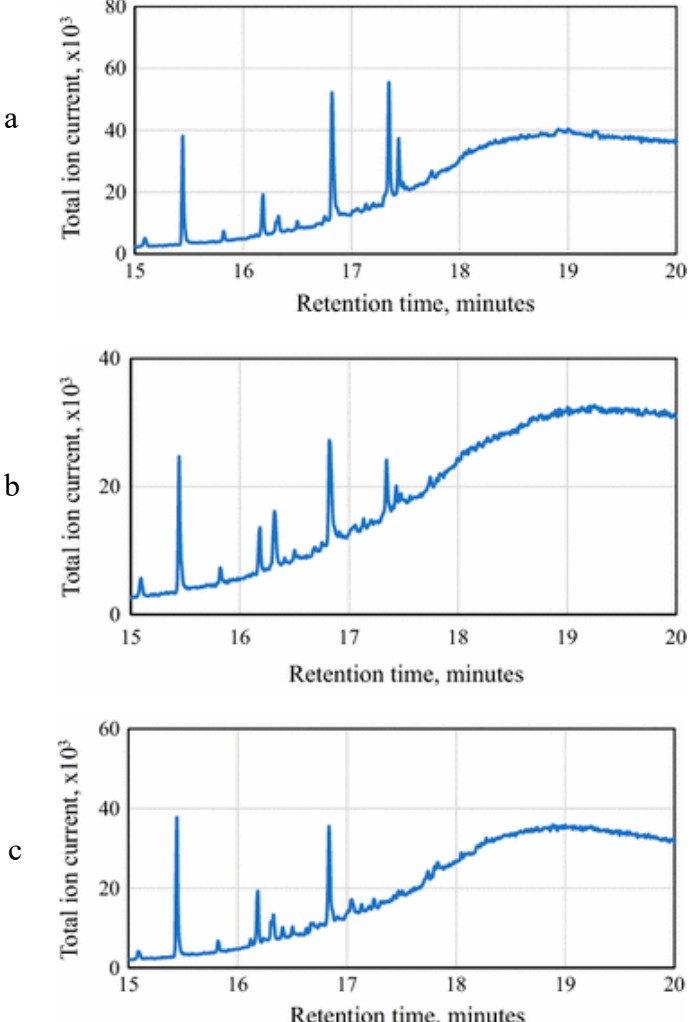

**Figure 13.** Total ion chromatogram for (**a**) unaged, (**b**) RTFOT-aged and (**c**) PAV-aged rejuvenated RAP (reprinted from Reference [61] with the permission of Elsevier).

### 4.5. Images Techniques as Useful Complementary Tools

Mokhtari et al. [62] have exploited the potential of FT-IR to investigate the effect of two different rejuvenators: a petroleum oil (called "A"), and a product derived from refined tall oil (called "B"). Rejuvenator "A" was added to 15% or 30% by weight on PAV, while the rejuvenator "B" was added to 10% or 20% by weight on PAV. These two dosage rates are the lower and upper limits of each rejuvenator type. Once the spectra were acquired, in order to determine if the use of the rejuvenators minimizes the oxidation, the $I_{co}$ and $I_{so}$ indices were calculated although the normalization seems to us to take place only over the C-H signal at 1459 cm$^{-1}$. As expected, PAV-aged samples show an increase in the $I_{so}$ value, as well as in the $I_{co}$ index.

On the contrary, by perusal of Figure 14 it can be seen that both rejuvenators at both concentrations show a significant decrease in $I_{so}$ values, while the decrease with respect to the $I_{co}$ index is moderate. However, both additives have proved to be effective in counteracting the oxidation of the carbonyl groups, as well as the sulfoxide groups. The Authors also thought to compare the FT-IR results with a Cryo-Scanning Electron Microscopy (Cryo-SEM) investigation for evaluating fracture surface properties of rejuvenator-restored through a digital image processing technique to quantify cracks developed on the fractured surface, due to the aging process. The choice of this technique comes from the Authors' consideration that the evaluation of the microstructural assessment of asphalt with SEM would not be confident, due to presence of volatile components in asphalt and its susceptibility

to electron beam damage. Instead, the use of Cryo attachment to SEM, implying a lowering of the samples' temperature well below the glass point, allows observing samples with greater beam and lower temperature sensitivity. They have acquired images with varying magnifications in different parts of the samples, to indicate the fracture surface characteristics of each sample. It is possible to notice that the asphalt sample has a rough and fractured surface with many fragments, due to the high stiffness caused by the aging process. Rejuvenator "A" makes the fracture surface of aged asphalt smoother with some remaining minor cracks or fragments. However, no further improvement of the surface texture can be observed when rejuvenator "A" amount was increased from 15 to 30%. Even adding rejuvenator "B" (10%) makes the fracture surface smoother, although at a minor extent, since significant amounts of fragments still holds. Instead, increasing the rejuvenator "B" amount from 10 to 20%, results in a significant improvement of the surface.

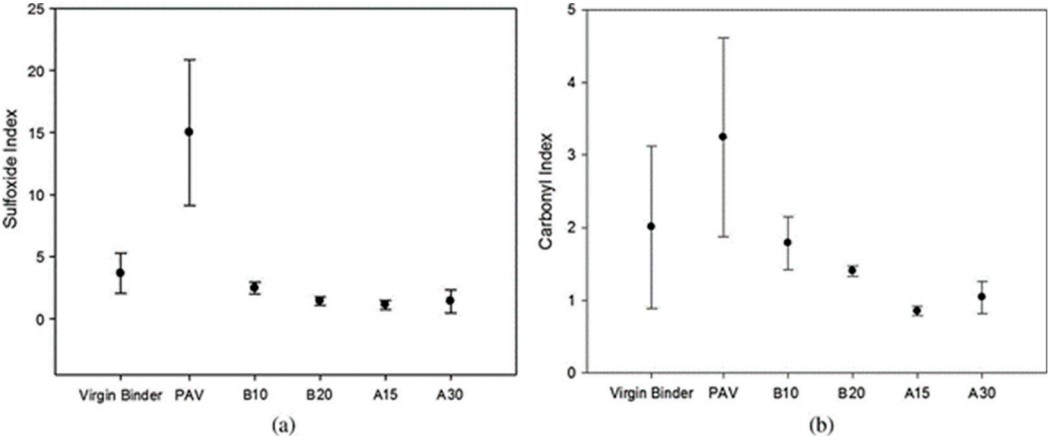

**Figure 14.** Calculated index values for various asphalt types (**a**) sulfoxide index, (**b**) carbonyl index (reprinted from Reference [62] with the permission of Elsevier).

An interesting aspect can be found in the work of Mokhtari et al., who notes that in order to identify the length and the gravity of the cracks, a digital analysis of the images was carried out using an edge detection technique. Briefly, various algorithms developed using MATLAB software DIP image toolbox were used to generate fracture models, including the crack propagation throughout the samples. Comparison with real samples allowed the selection of the best algorithm and a Fracture Index (F.I.) was defined in order to quantify the fracture capability of aged and restored asphalt samples.

As expected, as it can be seen from Figure 15, PAV-aged asphalt has the highest F.I., confirming more pronounced brittleness of the aged asphalt with respect to the aged asphalt with rejuvenators.

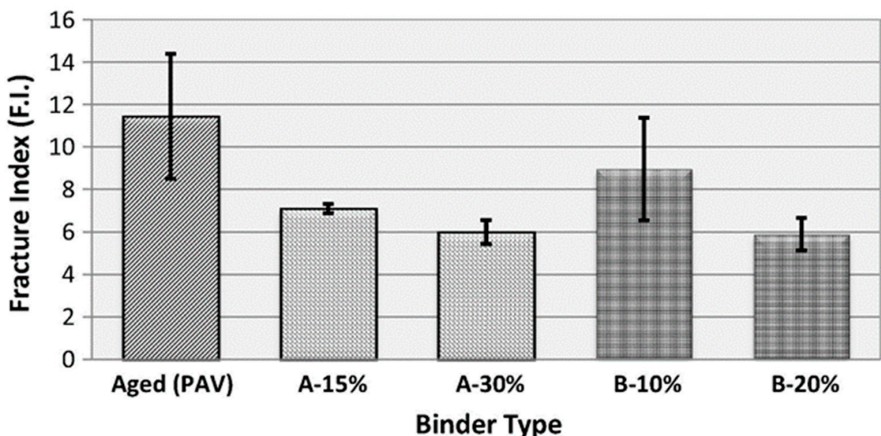

**Figure 15.** The comparative plot of fracture index for various asphalt types (reprinted from Reference [62] with the permission of Elsevier).

Figure 15 shows that the addition of 10% of the rejuvenator "B" could not significantly soften the aged asphalt. However, although 10% of the rejuvenator "B" was not completely effective in softening the PAV-aged asphalt, 20% of that rejuvenator shows the highest efficiency in preventing undesirable cracks at low temperatures. Moreover, both A-15% and A-30% asphalts were soft enough to prevent surface fractures.

Other authors have also focused their attention on the use of image techniques to better understand the structural changes induced by the use of rejuvenators. Indeed, Yu et al. [5] have used the AFM technique to analyze the materials' surface morphology. They have investigated two types of bitumen which are called by the Authors as asphalt binders, named AAD (PG 58–28) and ABD (PG 58–10), and two types of rejuvenators: one comes from fast and convenience food frying oil, here named WV oil, the other is an aromatic extract containing approximately 75% of aromatic oil and resin compounds with small amount of saturate oil. AFM images are shown in Figure 16.

The two virgin binders displayed different morphologies: virgin ABD has a dispersed phase with flake-like structures (with an average size of fewer than 2 μm in diameter) spreading over a smooth matrix continuous phase; virgin AAD, instead, clearly shows the elliptical domains with "bee-structures" (with axes of a few microns). The Authors attribute these differences to the chemical composition of the two samples, in particular, worth of note is the high wax content of the bitumen AAD (1.94%) with respect to the ABD bitumen (0.81%). Upon aging, in ABD, the size of the flake-like microstructures decreased while the quantity increased; on the other hand, in AAD, the "bee-structures" are still present with no obvious morphological changes. The addition of the aromatic extract in the aged ABD bitumen, instead, produces marked changes, giving elliptical domains with "bee-structures" at the middle of the domains. On the other hand, the use of the aromatic extract in the aged bitumen AAD gives smaller-sized "bee-structures". Even, the less noticeable contrast between the 'bee-structures' and the matrix suggests that the amplitude of the undulated "bee-structures" is smaller than that of the virgin and aged ones. On the contrary, the addition of WV oil in both aged binders do not produce a significant morphological modification.

The Authors conclude that the addition of the rejuvenator can cause big morphological changes, even bigger than those coming from the aging; however, this behavior is not general, depending on the specific rejuvenator. This because the asphalt binders' effect depends on its chemical behavior, and therefore, on the complicated molecular interactions which can establish with the other chemical species in the bitumen (and also the eventual inorganic particles). This is generally true for any additive. This conclusion is in accordance with the clues of a recent paper by Calandra et al. [1] which highlights the physical and chemical reasons for this. In this work, the Authors carried out a deep structural investigation by X-ray scattering on bare and additivated bitumen and found that asphaltene clusters hierarchically self-assemble to form aggregates at various levels of complexity, with different sizes up to the micrometer-sized domains dispersed in the maltene, and hold up by interactions of different strengths. The eventual presence of an additive triggers the formation of further intermolecular interaction in competition with those responsible for this self-assembly, causing a change of the size and shape of these aggregates.

The potential of the AFM has been exploited by Kuang et al. [23] to discriminate the effect of Dodecyl Benzene Sulfonic Acid (DBSA) as a solubilizer together with conventional rejuvenator (a blend of fluid catalytic cracking slurry—FCC—and bitumen with penetration of 70 grade) in two different aged (TFOT and PAV) bitumen.

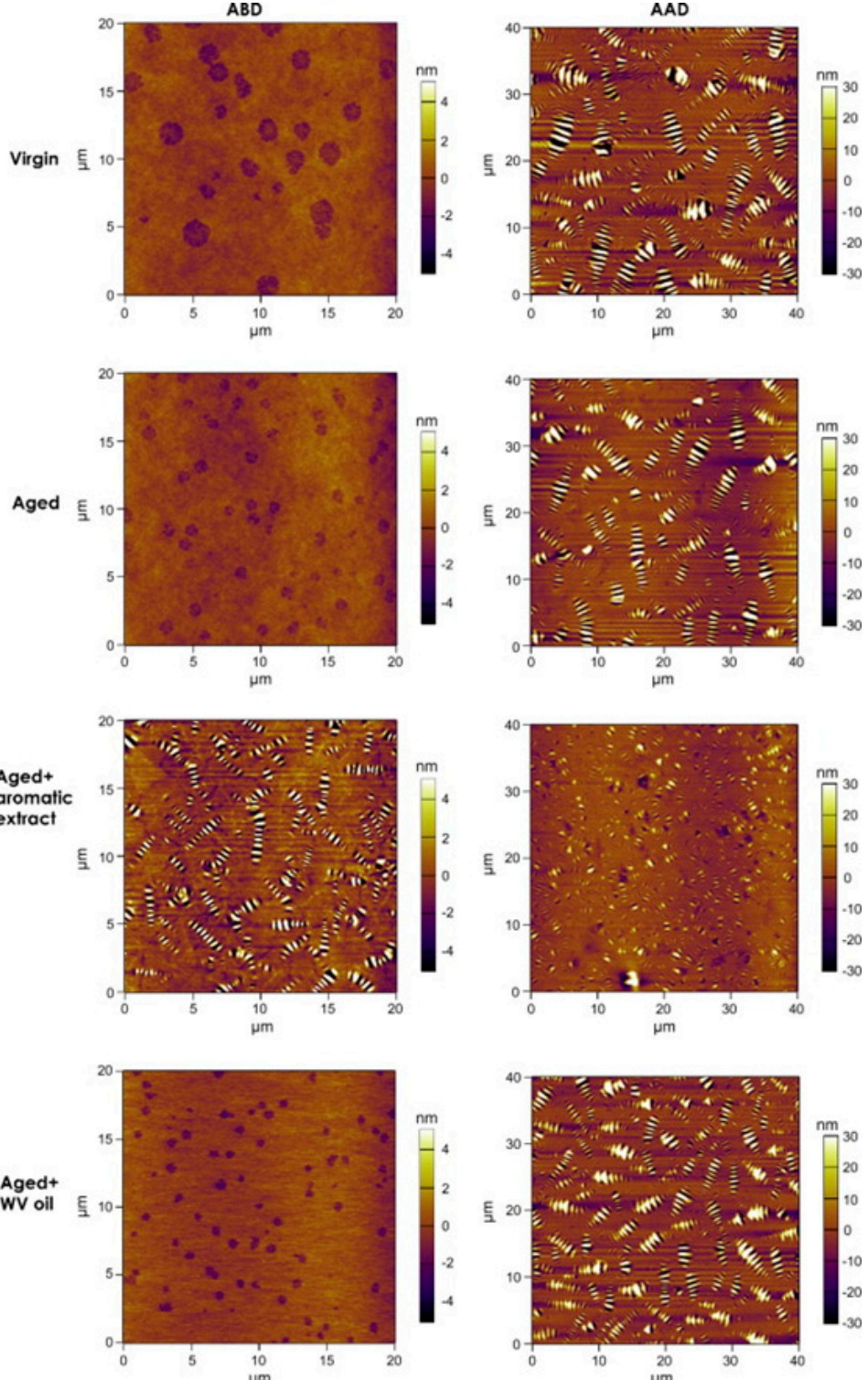

**Figure 16.** Topographic images of virgin (**top row**), aged (**2nd row**), aromatic extract (**3rd row**), and WV oil rejuvenated (**4th row**) ABD (left, $20 \times 20 \ \mu m^2$) and AAD (right, $40 \times 40 \ \mu m^2$) measured at room temperature (~20 °C). The colour scales range over 10 nm and 60 nm for ABD and AAD based samples, respectively (reprinted from Reference [5] with the permission of Elsevier).

Kuang et al. compared first the effect of aging on the virgin bitumen, whose clues are reported in Figure 17. This figure shows the AFM images of virgin bitumen, Thin Film Oven Test (TFOT) aged bitumen and PAV aged bitumen. The Authors have observed that the aging of the bitumen leads

to a change in the colloidal structure, and differently from Yu et al., an increase in the size of the bee-structures (see Figure 17). Then, they compared the effect of adding the conventional rejuvenating (CR) and the solubilised rejuvenating (SR) on the aged bitumen TFOT and PAV. The comparison in TFOT-aged bitumen and PAV-aged bitumen are reported in Figure 17. They found that in any case the surface of both TFOT and PAV bitumen become smoother with the introduction of CR or SR. However, in the case of TFOT there is no evident difference between the effect of CR and SR showing that, in this case, DBSA does not affect much the surface smoothing action exerted by the rejuvenator. On the other hand, an evident effect can be detected in the case of PAV. By inspection of Figure 18 (lower panel) and comparison with Figure 17, it is possible to notice how the addition of CR on the aged PAV bitumen does not bring marked morphological changes, just a slight reduction in the size of the bee-structures. On the contrary, the addition of SR helps asphaltenes of PAV aged bitumen to be re-dispersed, and this contributes to the performance improvement.

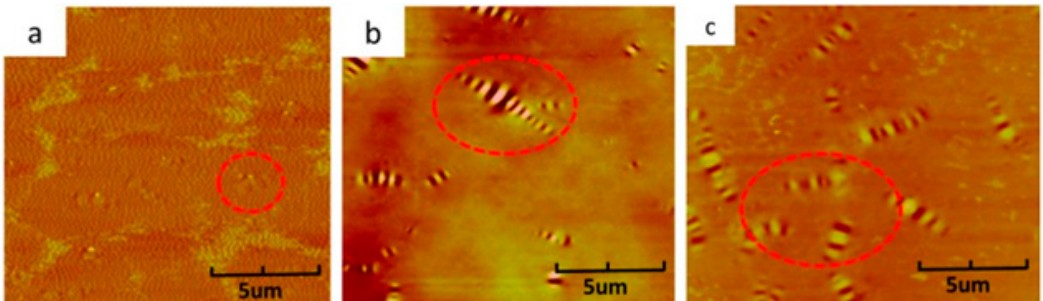

**Figure 17.** Atomic Force Microscopy (AFM) images of virgin bitumen, TFOT aged bitumen and PAV aged bitumen: (**a**) Virgin bitumen, (**b**) TFOT aged bitumen, (**c**) PAV aged bitumen (reprinted from Reference [23] with the permission of MDPI open access journal).

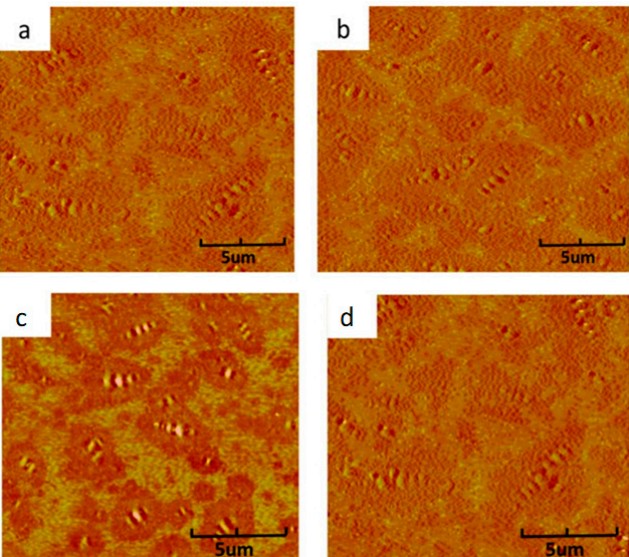

**Figure 18.** Upper panel: (**a**) Regenerated TFOT aged bitumen with 10 wt% CR, (**b**) regenerated TFOT aged bitumen with 10 wt% SR; lower panel: (**c**) Regenerated PAV aged bitumen with 10 wt% CR, (**d**) regenerated PAV aged bitumen with 10 wt% SR (reprinted from Reference [23] with the permission of MDPI open access journal).

For the Authors DBSA (1.5% by weight of rejuvenator) can react with asphaltenes via its sulfonic group to form a solvation layer covering on the surface of asphaltenes clusters, thus, interfering with the colloidal structure transformation of aged bitumen. Therefore, PAV aged bitumen can be recovered

from Gel to Sol-Gel by 10 wt% SR, and the dimension of a bee-like structure formed by asphaltenes can be approximately that of virgin bitumen.

Also, Nahar et al. [63] have explored the potentiality of AFM techniques to investigate the effect of rejuvenator on the aged bitumen. They used a bitumen aged through Rotational Cylinder Ageing Tester (RCAT) for testing two types of rejuvenators, namely, BM1 and CM1. They analyzed the pristine, aged and doped aged bitumen by means rheology and AFM.

From the rheological analysis, they were able to observe that the aged bitumen P1 shows higher complex shear modulus and lower phase angle compared to the pristine bitumen. Obviously, ageing makes the bitumen stiffer (Figure 19a) and less viscous (Figure 19b). The neat rejuvenators show very distinct behaviors. In fact, the rheology of the BM1 rejuvenator shows a lower viscosity compared to bitumen, while CM1 rejuvenator has different rheological characteristics. Indeed, it displays a much lower shear modulus at low frequencies. There is even a behavior of dilatant fluid or shear thickening at a frequency of about 3–5 Hz. The Authors thought that this was due to the presence of suspended particles like structures in rejuvenator CM1 or the formation of such structures at higher shear rates. However, the addition of rejuvenator BM1 into the aged bitumen causes a decreasing of the complex shear modulus, while the phase angle increases to the value of the virgin bitumen. CM1 rejuvenator on the aged bitumen leads to a lower complex shear modulus than pristine bitumen for both the concentrations tested. On the contrary, as expected, the phase angle is almost equal to (10% *w/w* CM1) or higher than the pristine bitumen (25% *w/w* CM1). It is worth noting that in a blend with aged bitumen any signature of the dilatant nature of the pure rejuvenator CM1 is completely lost.

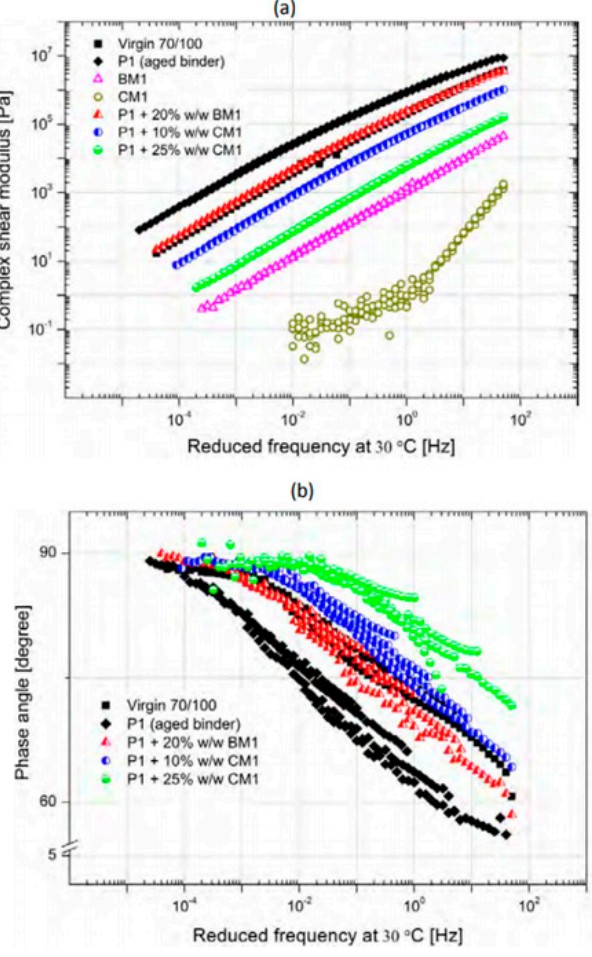

**Figure 19.** (**a**) Complex shear modulus of pristine, aged, doped aged and rejuvenator; (**b**) phase angle of pristine, aged and doped aged bitumen (reprinted from Reference [63] with the permission of SAGE Publications).

In our opinion, the AFM analysis conducted by Nahar et al. is commendable. To our eyes, it seems to be almost the most complete job among all the AFM-based works we have examined. In fact, the authors were not limited to the identification of domains and bee structures, having identified new types of structures (e.g., tertiary and quaternary) providing an accurate description. Figure 20(ai) shows the AFM phase image for the pristine bitumen. It is possible to estimate the length of the domains long axes which falls into the range 2–6 μm. The topography images show "wrinkling" in the middle of the domain. Finally, the domains result to be buried about 2–5 nm with respect to the average height of the continuous phase. On the contrary, the aged bitumen is very different from the pristine one. Figure 21 shows the same elliptical domain (i) and matrix (ii) of pristine bitumen. Moreover, it is possible to observe the tertiary phase, which consists of fine dark arcs and spots dispersed throughout the matrix (iii). This tertiary phase displays the lowest phase shift, and it is the softest phase. From the topography (shown in Figure 20 (ii)), it turns out that the elliptical domains are 5–8 nm lower (buried) the matrix surface, on average, while the tertiary phase protrudes above the matrix phase by 3–5 nm.

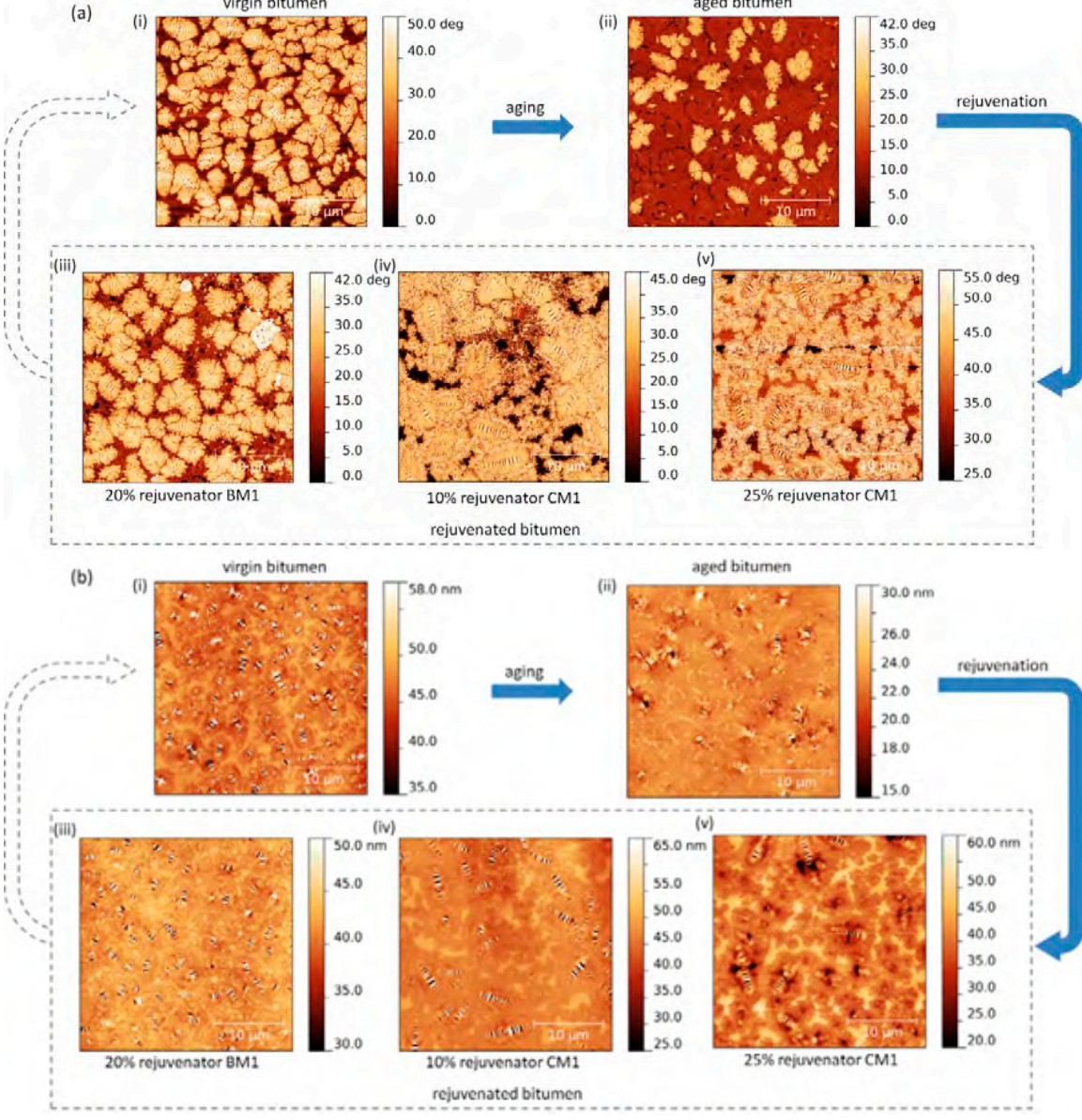

**Figure 20.** AFM (**a**) phase images, (**b**) topography images of pristine, aged and doped aged bitumen (reprinted from Reference [63] with the permission of SAGE Publications).

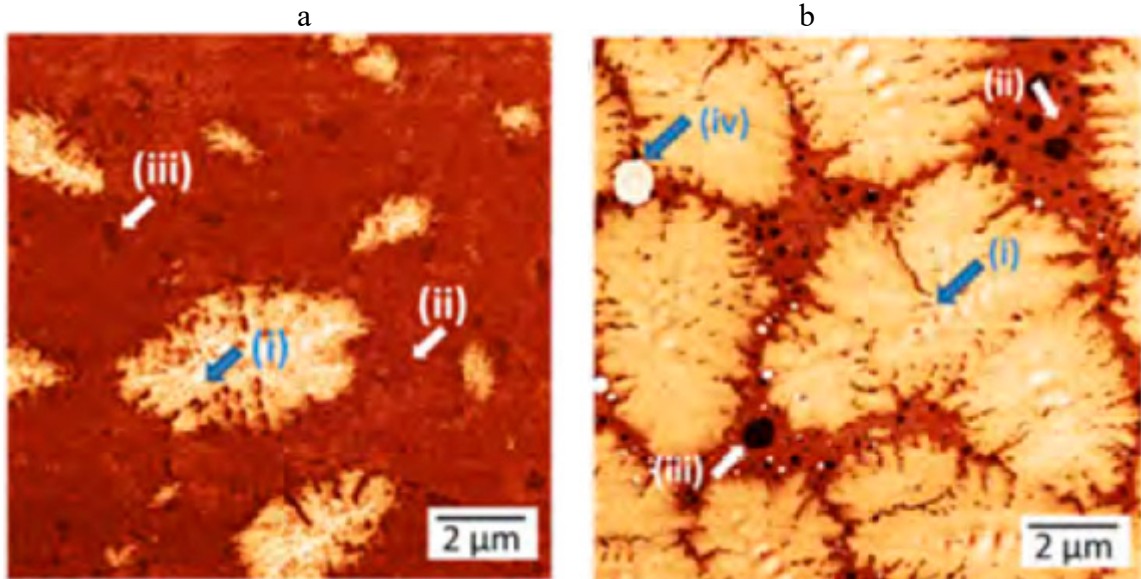

**Figure 21.** AFM phase image of aged bitumen (**a**) and aged bitumen doped with 20% of BM1 (**b**) with (i) elliptical domains, (ii) matrix, (iii) tertiary phase and (iv) quaternary phase (adapted from Reference [63]).

From Figure 20(aiii), it is possible to notice that the addition of 20% of BM1 to aged bitumen restores the microstructure. However, now it is possible to distinguish a new phase at the boundary of the domains and at the interstitial spaces of consecutive domains. A new quaternary phase is found, see Figure 21(biv). This new phase has the highest phase shift (highest stiffness), and it appears in almost circular shapes with sizes in the range 15.2–4 μm.

As it can be seen from Figure 20 (iv) and (v), the addition of the CM1 additive at both concentrations causes a change in morphology: needle-like particles, 20–90 nm wide and 50–250 nm long, appear. The network becomes the main phase at a microstructural level at the increasing of CM1 concentration.

In order to understand if the microstructure evolves with time, the Authors analyzed the samples, again, after seven days. The aged bitumen and the aged doped with 20% BM1 did not show any change over time. On the contrary, the aged bitumen doped with 10 and 25% of CM1 evolve over time (see Figure 22). The biggest change can be observed for the blend with the lower (10%) concentration of CM1 rejuvenator (see Figure 22(aii)). Over time, some effects can be highlighted: needles are expelled from the matrix phase, disconnected domains only comprising the network phase are born, and the wrinkling in the elliptical domains tends to decrease. The network phase, in this period, has formed a kind of bilayer around the elliptical domains, consisting of 200–300 nm stiffer layer (higher phase shift, light colour in Figure 22(bii)), surrounded by a 1 μm somewhat softer layer. Both layers display the typical pattern of randomly oriented needles, typical of the network phase. The higher stiffness layer may be just a denser packed region of the network phase. On the contrary, for the higher concentration of CM1 rejuvenator (25%) the effect of the time on microstructure is less obvious. Indeed, it is possible to notice islands solely consisting of the needle network, as well as the bilayer of 15 needles surrounding the elliptical domains. It is noticeable that, here, the stiffer (white) region of the needle network phase is the more prominent phase. Also, at a 25% concentration of CM1, the wrinkling in the elliptical domains remains over time, though the oscillation amplitude has decreased.

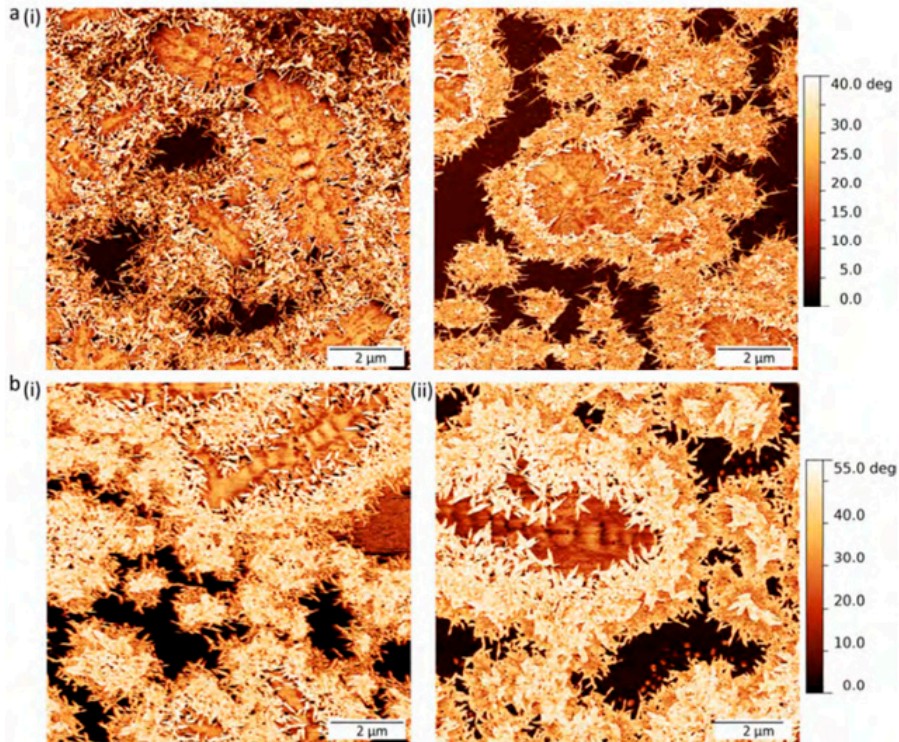

**Figure 22.** Time evolution of microstructure of aged bitumen doped with CM1. AFM phase images; (**a**) 10% CM1 and (**b**) 25% CM1, (i) after preparation (ii) after seven days (reprinted from Reference [63] with the permission of SAGE Publications).

Kuang et al. [64] evaluated the effect of composite rejuvenator in comparison with a common rejuvenator by means of dynamic shear rheometer (DSR) and atomic force microscopy (AFM). The composite rejuvenator, named RRA, was laboratory-prepared by blending the light weight oil rich in aromatics with a chemical compound containing polar group. The common rejuvenator, denoted as CRA, was also prepared in a laboratory. They used for these researches the bitumen SK-70, whose physical properties are listed in Table 2.

**Table 2.** Physical properties of SK-70 bitumen (reprinted from Reference [64] with the permission of Springer Nature).

| Index | Value |
|---|---|
| Penetration (25 °C, dmm) | 73 |
| Ductility (15 °C, cm) | >150 |
| Softening point/°C | 45.2 |
| Viscosity (135 °C, Pa·s) | 0.6 |

The aged binder was prepared through the aging of SK-70 by Thin Film Oven Test (TFOT). Aged bitumen was doped with 4; 6; 8 and 10 wt% of the two kinds of rejuvenator.

The results are shown in Figure 23: the composite rejuvenator (RRA) has a greater effect with respect to the common rejuvenator (CRA). When the content of RRA is 10 wt%, the values of penetration, ductility, softening point and viscosity are very close to the values of pristine SK-70. Therefore, the Authors believe that the rejuvenator RRA is able to restore the colloidal structure by increasing the aromatics content. In addition, the polar groups of RRA can react with the asphaltene in aged bitumen, decreasing the asphaltenes content themselves. On the contrary, according to them, chemical reactions between the common rejuvenator and asphaltenes molecules do not take place, due to the lack of polar groups in common rejuvenator.

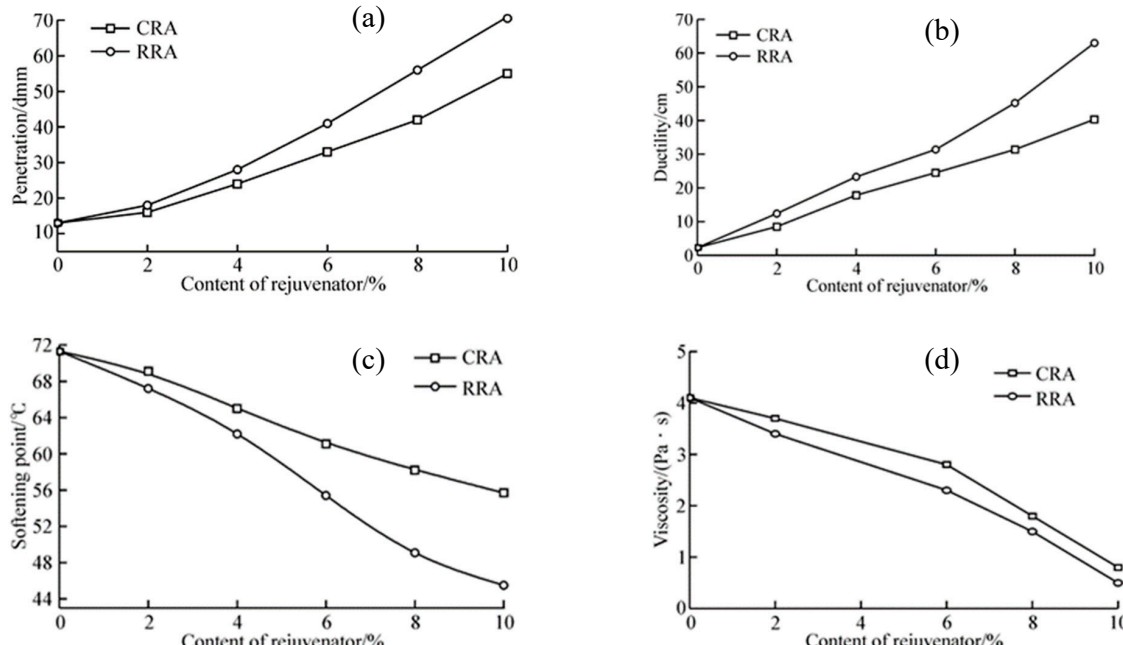

**Figure 23.** Effect of rejuvenators on penetration (**a**), ductility (**b**), softening point (**c**) and viscosity (**d**) (reprinted from Reference [64] with the permission of Springer Nature).

From the rheological analysis, it results that the rutting factor, shown in Figure 24a as a function of temperature, of the aged binder is greater than the virgin one. The addition of 10 wt% of RRA and CRA causes a decrease in the rutting factor, but sample with RRA gives a trend similar to the pristine one. Obviously, the aged bitumen is very brittle and easy to crack under load at low temperature, in fact, has a high fatigue factor (see Figure 24b) compared to virgin bitumen. The poor regenerative power of CRA is evident. Aged asphalt with 10 wt% CRA and aged asphalt shows similar fatigue factor curves. Nevertheless, the addition of 10 wt% of RRA improves the fatigue resistance considerably, being the fatigue factor trend close to that of virgin bitumen. In order to evaluate the effect of rejuvenators on the bitumen structure, the Authors carried out AFM analysis. From topographic images, the virgin asphalt (see Figure 25a) seems to be rather smooth; instead, the aged asphalt, (see Figure 25b) appears to be more wrinkled. Even, the surface of aged asphalt doped with 10 wt% of CRA seems to be rougher with flocculated structure just like that of the aged asphalt, see Figure 25c. The addition of RRA restores a smooth surface like that of virgin asphalt, Figure 25d. The disappearance of the flocculated structure is due to the dissolution of asphaltenes. Therefore, it can be concluded that aged asphalt can experience good recover of its original microstructure by means of the composite rejuvenator: an effective rejuvenation of the aged asphalt performance can be claimed.

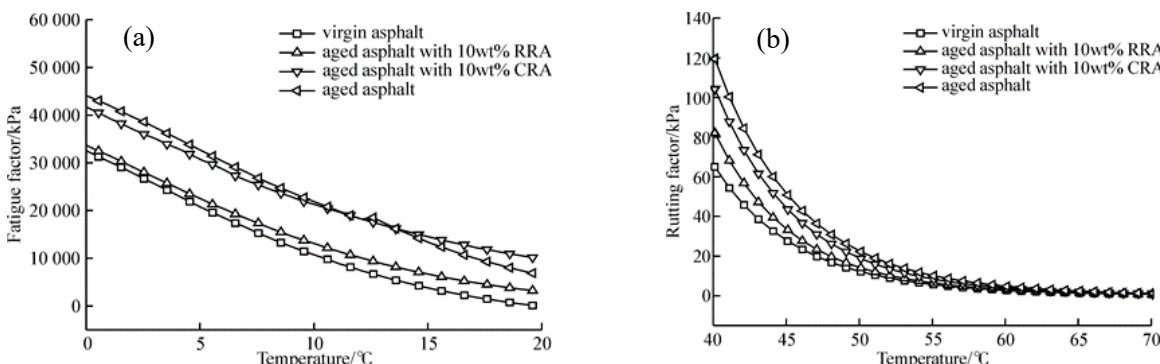

**Figure 24.** Effect of the rejuvenators on the Rutting factor (**a**) and on the Fatigue factor (**b**) (reprinted from Reference [64] with the permission of Springer Nature).

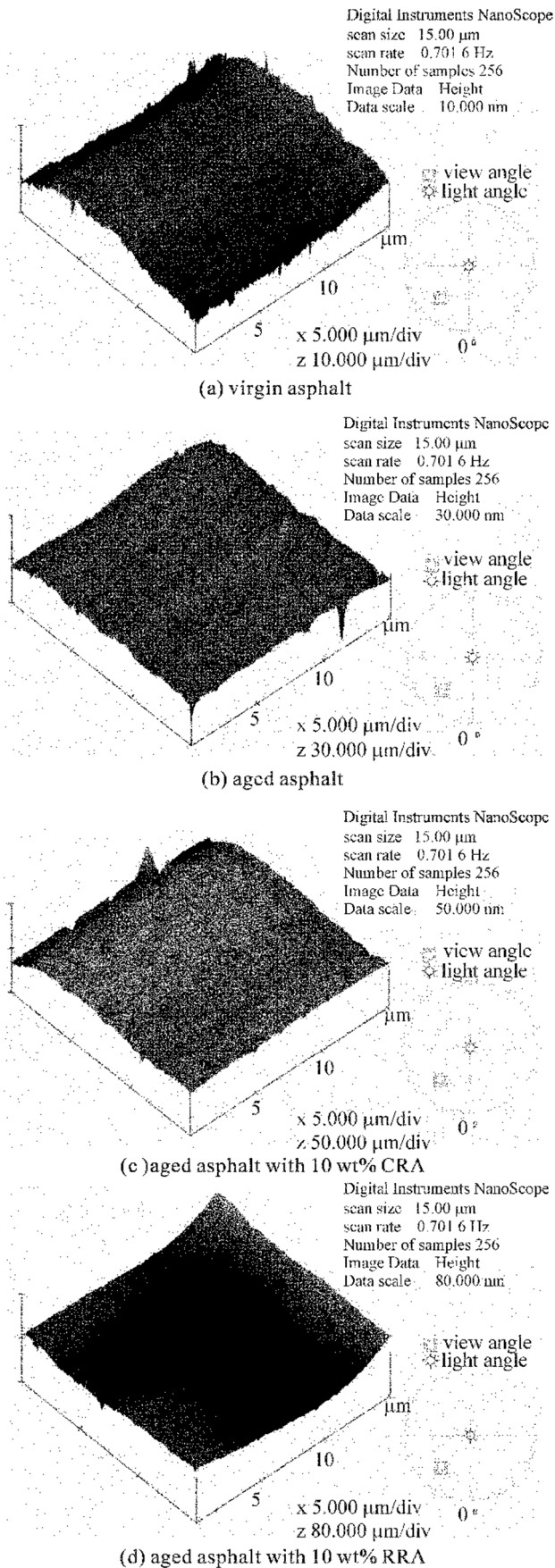

**Figure 25.** Topographic AFM images of samples (reprinted from Reference [64] with the permission of Springer Nature).

## 5. Perspectives

### 5.1. Improving Rejuvenators Characteristics

As already discussed above, a rejuvenator is commonly made by simple oil waste for their low-cost and obvious environmental concerns. In fact, oil waste is the by-product of edible oils surely more produced in the world. Therefore, its re-utilization may provide a feasible method to minimize the amount of generated waste, providing a positive environmental impact. From a chemical point of view, the oil subjected to elevated temperature changes its chemical composition, producing molecules with high anti-oxidant properties, which are certainly advantageous, for example, in the exhausted oil utilization as rejuvenating of bitumen. However, the waste oils have some disadvantageous, such as poor low temperature fluid properties, a propensity to oxidative degradation, a susceptibility to hydrolysis in acid media, which limits its application. For this reason, it is actually of greatest interest the proposal of chemical modifications of their structure to improve its physicochemical properties. In particular, if search should be narrowed only to vegetable oil waste (i.e., Waste Cooking Oils (WCO), then soybean oil, canola oil, coconut oil, castor oil, etc.), esterification, [65] hydrogenation [66], epoxidation [67], acylation [24] are only some examples of adopted chemical transformations to enhance the performance of these oils. In fact, the major components of vegetable oil waste are triglycerides esters of glycerol with saturated and unsaturated long-chain fatty acids, therefore, the transesterification reaction may represent the most valid method to change their properties, and it progresses through hydrolysis and successive esterification of the hydrolyzed products as schematically shown in Scheme 1.

$$\text{A)} \quad (RCO_2)_3C_3H_5 + 3H_2O \rightleftharpoons 3\,RCO_2H + C_3H_8O_3$$

$$\text{B)} \quad 3RCO_2H + 3ROH \rightleftharpoons 3\,RCO_2R + 3H_2O$$

R = Saturated or unsaturated long carbon-chain

**Scheme 1.** (A) Hydrolysis reaction to form long-chain fatty acids; (B) esterification reaction to produce esters of long carbon-chain fatty acids.

This procedure paves the way to subsequent chemical modifications. An example is reported from Xiang et al. [68] that carried out the transesterification reaction by the treatment of exhausted oils with NaOH in $CH_3OH$ to obtain methyl esters of fatty acids. The latter were transformed in variously substituted triester derivatives by the first epoxidation of unsaturated bounds present on of carbon backbone of R and a subsequent opening of oxirane ring to produce a final matrix with increased physicochemical properties, (i.e., better anti-wear ability, improved oxidation stability, etc.). However, it is evident that in proposing new synthetic methodologies of structural transformations, the chemists should take into account cheap and eco-friendly procedures. For this reason new perspectives on this typology of reaction may be, for example, the introduction of low-cost catalysts (i.e., Lanthanide salts, Fe salts, Zn salts, Cu salts, etc.), which are known to be employed in many chemical manipulations [69,70] in combination with green solvents, such as ionic liquid, Deep Eutectic Solvents (generally named DES) or water or Microwave irradiation [71,72]. For example, in order to transform oil waste into more performing products, it is possible to suggest a synthesis process to open oxirane rings, formed on the unsaturated bounds of R chain, in mild conditions, introducing transformable functional groups, such as nitriles or nitro that may be subsequently oxidized in environmentally friendly solvents. Another suggested modification may be the introduction of acyl groups with alkyl or aromatic chains on the unsaturated scaffold of the carbon-chain [73–75]. The presence of alkyl or aromatic groups could lead to forming secondary interaction, such as Wan der Waals or $\pi$-$\pi$ interactions that may confer to the modified oils a major branched substructure, varying the physicochemical characteristics, increasing the chemical stability of the substrates and obtaining their better performance.

An alternative to the chemical catalysts could be the enzymatic catalysis, as proposed by Avisha and co-workers [76]. Candida rugosa lipase was used to hydrolyze oil wastes in a water solution with successive esterification by Amberlyst 15(H) resin catalyst. In addition, orange lipase deriving from waste was used from Okino-Delgado et al. in transesterification reaction, realizing remediated oils with higher performance profile [77]. However, the enzymatic catalysis through the commercial and the homemade lipase is considerably less convenient than the chemical one, because of higher reaction costs, specific reaction conditions, less manageability, the greater facility of degradation. On the other hand, the negative impact of oil waste on the environment and humans directs the scientists to propose its re-use to reduce the produced amount. Then, the chemical manipulation of the exhausted oils fits perfectly in the contest of its recycling. Obviously, it is important to always make a cost/benefit ratio. In fact, it is correct to think that chemical modification of an oil matrix has a major cost than its use as such, but it is equally true that with a view to having products, such as bitumen rejuvenating ones with enhanced physicochemical properties, the future of research in this field is to invest in rebirth of oil waste by innovative synthetic methods.

*5.2. New/Novel Rejuvenators*

The overall characteristics of a bitumen are, as a matter of fact, the consequence of its complex structure. The term "complex" does not refer to "complicated" or "hard to describe" but, instead, the term is used on a physical basis. The peculiar aspect involved in this topic will be better clarified in the next paragraph. Aging perturbs and changes the complex organization of a bitumen through various mechanisms whose consequence is to change the overall complex organization of the material and not, strictly speaking, one single specific aspect of the molecular organization of the bitumen. That is why, if only one specific structural feature is looked at, it may not correlate the dynamic behavior. In this respect, novel rejuvenators can be thought of. A typical class of molecule usually related to complex behavior is that of surfactant, and more generally that of amphiphiles. Amphiphiles, simultaneously possessing polar and apolar moieties within their molecular architecture, can give a wide scenario of possible intermolecular interactions: polar–polar, polar–apolar, apolar–apolar interactions, eventual directional H-bonds, steric hindrance, etc. For this reason, some of them are a surfactant, i.e., surface-active agents, when dissolved in water: they expose to the air their apolar part while binding water through their polar head, thus, decreasing the surface tension [78,79]. The same principle holds when trying to mix polar and apolar substances. Bypassing their natural tendency to remain separated, they actually can be effectively mixed if an amphiphile is present. Thanks to its capability to simultaneously linking both the polar and the apolar phases, the amphiphile act as a bridging molecule between the two. The two phases can be, therefore, homogenized to such an extent that the system can become homogeneous at the macro-scale although heterogeneous at the nano-scale, with the formation of local domain of one phase stabilized by one or more layers of opportunely oriented amphiphilic molecules and dispersed in the other phase [80]. Micelles, vesicles, bicontinuous structures and liquid crystals are only examples of stabilization of the systems through the formation of local intermolecular assemblies. This principle can be used, in our opinion, to the bitumen cases also. Let us consider that, in the micellar model, the bitumen is constituted by polar aggregates stabilized by polar resins and dispersed in a more apolar matrix. In this case, an amphiphilic molecule can bind on a side the asphaltene cluster, and on the other side, the apolar maltene phase. The overall outcome of these simultaneous interactions would be to disperse the asphaltene clusters contrasting aging better, or even, drawing back to rejuvenation. On the other hand, it must be admitted that the general mechanism of action shown by amphiphiles is already well-known and used for several issues in various fields: amphiphiles have been proved to be effective in stabilizing organic molecule clusters within an apolar solvent [81,82], as well as metal clusters [83], nanoparticles [84], and ionic clusters [85], so in our opinion their direct application to bitumen represents an obvious and immediate step. This idea has been recently tested in preliminary works where the surfactants have been successfully used to prepare warm mix asphalt binders. The results showed that the use of the

surfactant-based additive reduces surface free energy. It increases after short-term (Rolling Thin Film Oven) and reduces after long-term (Pressure Aging Vessel) aging [86]. Moreover, the addition of DBSA (Dodecylbenzosulfonic acid) based surfactant enhanced viscoelastic response of bitumen and reduced glass transition temperatures since it promotes the association of asphaltene molecules/aggregates into larger clusters in bitumen [87].

## 6. Forefront/Vanguard Techniques Facing Complexity in Bitumen

### 6.1. Complexity

The aim of this last paragraph is to furnish some hints on the future developments of new techniques for the investigation of bitumen. To do so, it is advisable first to clarify the exact nature of such systems, so it is needed to shed light on what is meant with the concept of "complex systems". Complexity is based on a hierarchical relationship between constituents and objects. Just to introduce the topic by an example, elementary particles are somehow assembled to form atoms, atoms are assembled to form molecules, molecules can assemble to form living cell, opportunely organized living cells can constitute tissues, and insisting with such a kind of reasoning, tissues, organs, human beings, society, etc., can be consecutively considered in an escalation within the principle a high number of levels. What is called "constituents" can be assembled together to form a complex object which, in turn, can be one of the "constituents" making an even bigger (i.e., more complex) object. So, what are called "constituents" belong to a specific "level", but when they are assembled to form a "bigger" object, a successive level is reached. These are what are called levels of complexity, and it can be misleading to deal with "bigger" or "smaller" systems, because it is not a matter of size: it is correct to deal with different levels of complexity. The peculiar features possessed by the elements belonging to each level of complexity are the consequence of novel emerging and unexpected properties that can arise when passing from a level to the successive. Since constituents are interacting, in fact, the complex system is not the mere collection of its building blocks so that the overall properties cannot be obtained by simple extrapolation of the characteristics of their constituents. Interestingly, novel and unexpected emerging properties can arise when passing from a level to the successive. Complex materials exhibit, therefore, spatial correlations between their constituents at different scales.

### 6.2. Probing Complexity

Bitumen is certainly a complex system, due to the asphaltene aggregation taking place at different length-scales, and due to the specific molecular aggregation involving different chemical species (see Sections 1 and 2). As in micellar systems, there are structures living for milliseconds (micelles) which can also be spatially correlated at high concentrations, in a similar way the same can be expected for bitumen. The term "structure", obviously crucial for an adequate description of a complex system is to be intended as strictly related to the spatial correlation between the constituents of the system. On the other hand, it cannot be neglected that for applicative purposes, the study of material must advisably involve non-invasive techniques. This is exactly what scattering techniques probe. The structure factor is inherently contained in the output of a scattering experiment. Scattering techniques not only give direct information on the structure possessed by the system, but also give its synthetic fingerprint. Such information (characterization of the constituents and their correlation to longer scales) are crucial in order to establish the proper connection between nanoscale morphology and bulk properties in complex systems. Here we want to emphasize that the "structure" involved in complexity is exactly defined in terms of the direct observable through scattering techniques, i.e., the existence of preferred distances (spatial correlations) among specific constituents of the system and taking place at different length-scales [88] so will give, in the following paragraphs, some hints on the theoretical background at the basis of this technique. Of course, other methods, especially microscopy-based, are available, i.e., Atomic Force Microscopy (AFM), Scanning Electron Microscopy (SEM) and Fluorescence Microscopy were used to investigate bitumen doped by polymers. The AFM and SEM have been used in order to

study the structure of asphalt while Fluorescence Microscopy was used to aid in understanding the structural changes occurring when polymers are added to the asphalt. The Atomic Force Microscopy was able to study the only structures of the asphaltenes. On the contrary, the Fluorescence Microscopy can only reveal the presence of fluorescing molecules. Oil exhibits autofluorescence when irradiated with shorter wavelength light, such as UV light, but for bitumen, there is very little fluorescent light emission because the oil phase is mixed with an asphaltene and a resin phase which do not exhibit any autofluorescence [89]. Performances and rheological behavior of a bitumen, however, is also a consequence of the dynamics involved at the molecular basis. Whereas, the above-cited techniques essentially probe the structure, a vanguard method to probe the dynamics fit for accurate bitumen characterization must also be individuated. Therefore, after a presentation of the scattering techniques, it will be shown another vanguard technology that can be used to deeply analyze the bitumen dynamics: the Relaxometry NMR, consisting in the measurement of the $^{1}$H-NMR relaxation times of bitumen at low magnetic fields.

### 6.2.1. Scattering Theory

Apart from the development of scattering techniques at the large-scale facilities, recent improvements in lab instrumentation and the related beam intensities have greatly enhanced the importance of scattering methods in the structural characterization of complex materials. The quality of the recorded spectra is becoming adequate to extract information even from complex systems as bitumen is. The fundamentals of scattering techniques are, therefore, now given with the only scope to show how they are able to furnish the synthetic view of the material structure [90] of any nature [88]. A typical scattering geometry is reported in Figure 26: an incident (monochromatic or monochromatized) beam from a given source (e.g., Neutrons, X-rays, visible light, electrons, etc.) with incident wavevector ($k_0$) impinges on the material system under investigation. The scattered radiation intensity *I* is collected by a detector at a given scattering angle $2\theta$ with respect to the incident radiation direction. The difference between the scattered ($k_F$) and incident ($k_0$) wavevectors furnishes the scattering wavevector $q = |k_F - k_0| = (4\pi n/\lambda)\sin(\theta)$ (where n is the index of refraction of the medium and $\lambda$ is the wavelength of the employed radiation). For light n = 1.33 for a water medium, whereas, for X-rays and neutron n is very close to unit. It is worth noticing that a scattering experiment furnishes information over distances that are of the same order, or bigger, than the wavelength $\lambda$ of the source radiation. The scattering is generally described as arising from the constructive interference coming from objects that are embedded in a continuum medium (treated as constant background). The interaction of radiations with materials is characterized then by a scattering length $b_i$, and its scattering length density (SLD) which is given by $\rho(r) = \sum_i \rho_i(r)b_i$ where $\rho_i(r)$ is the local density of scatters of type *i* [91,92].

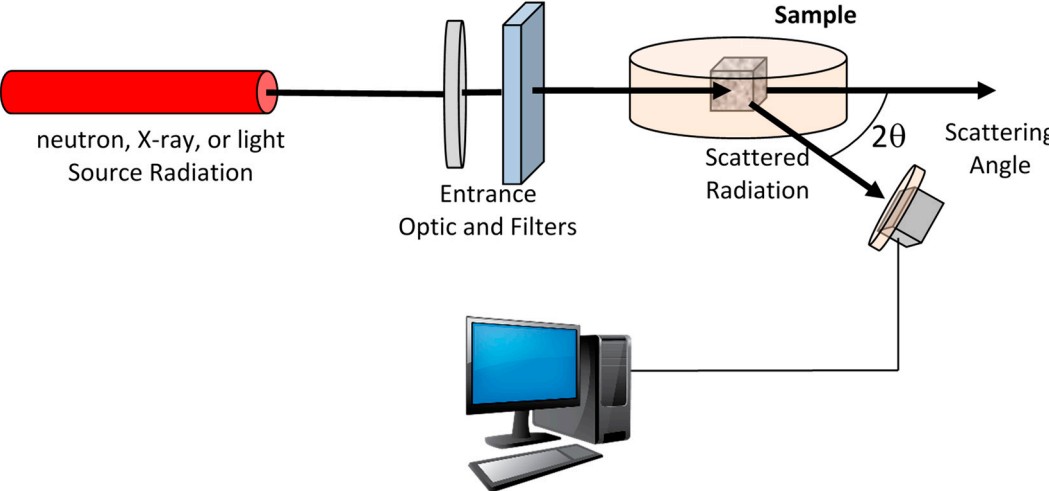

**Figure 26.** Schematic setup of a scattering experiment.

So, for example:

(i)   neutron scattering arises through (short-range) nuclear interactions (or magnetically, if atoms have unpaired electron spins), while the scattering length depends on the nature of the nuclei of the reference atoms.

(ii)  X-rays scattering comes from the interactions among all the electrons in the material under investigation. In this case, the scattering density can be traced back to the electron density.

(iii) In the case of Light photons, which have lower energy than X-rays ones, are scattered only by the outer part of the electronic cloud of an atom. In this case, the scattering length density is proportional to the polarizability of the materials.

Of course, different radiation (particle) sources give different sub-techniques. However, in the following, the different information that can be derived from the various scattering angle ranges, which in turn give different methods of analysis, will be shown.

6.2.2. Scattering of Neutrons (SANS), X-rays (SAXS) and Light

Scattering probes the statistical ensemble of the nano-structures and deals with the diffusion of electromagnetic (or particle) waves by heterogeneities in material systems [93]. At small angles 2θ, the (coherent) scattering intensity in the so-called "static approximation" is given by:

$$I(q) \propto \left[ \sum_i b_i\, e^{iqR} \right]^2 \qquad (2)$$

where $b_i$ is the scattering length of the particle (chemical species) that occupy the position $R$ in the material system. For SANS experiments the scattering length density $\rho(r)$ of the sample is defined as $\rho(r) = \sum_i n_i(r) b_i$, where $b_i$ is the scattering length of the nucleus of type $i$, while $n_i(r)$ is the corresponding number density of such nuclei. For X-ray scattering $\rho(r) = \left( \frac{e^2}{mc^2} \right) n_{el}(r)$, where $\left( \frac{e^2}{mc^2} \right)$ is the Thompson scattering length of the electron, and $n_{el}(r)$ is the electron number density.

By replacing $b_i$ by a locally averaged scattering length density $\rho_i(r)$, (where $r$ is a variable position vector), it is possible to perform an integration over the sample volume, V:

$$I(q) \propto \left[ \int_V \rho(r) e^{iqR} d^3 r \right]^2 \qquad (3)$$

If isotropic samples are considered (i.e., where the orientation effects are averaged, due to the radial symmetry), the scattering intensity *I(q)* can be expressed as:

$$I(q) \propto \int_0^\infty (\rho(r))^2 \frac{\sin qr}{qr} 4\pi r^2 dr \qquad (4)$$

Very often, it is not immediate from the experimental Small Angle Scattering (SAS) intensity profile to obtaining direct information about the $\rho(r)$ function by inverse transform methods. In this case, SAS interpretation is based on the choice of suitable models expressed in terms of specified functions, which are capable of furnishing information on specific parameters connected to particular properties of the material system under consideration. The name of the technique is then further characterized by the probe which is used: if the probe is made of X-rays, then SAXS is considered; if the probe is a flux of neutrons, then SANS holds, etc. If the source is light, then it will obviously deal with light scattering. However, it must be noticed that, when dealing with light scattering, another technique, called dynamic light scattering (DLS) also known as photon correlation spectroscopy (PCS), is usually referred to. In such a method, time fluctuations of the scattering intensity as a consequence

of the Brownian motion of nano-scatters in a solution are recorded. Then, a time-dependent scattering function is derived to the diffusion coefficient of the particles (the scatters) dispersed in the liquid phase [94,95]. In contrast to this (more widely known) technique, the static light scattering (SLS) configuration resembles the typical scattering apparatus.

In the case of wide-angle scattering, higher *q* values are considered, which means that shorter distances are explored. In a typical Wide-Angle Scattering experiment, which can use in principle the same sources as small-angle, usually, the scatters are the atoms themselves. Typical interatomic distances are, therefore, probed: in the case of pure crystals, where positional order is dominant, this gives the famous Bragg law:

$$2d \, \sin\theta = n\lambda \tag{5}$$

where *n* is an integer and *d* is the characteristic distance of reticular planes.

However, when the order is weak, the principle still holds, and wide-angle scattering can also be used for amorphous materials as in the case of bitumen. In this case, typical interatomic distances are unveiled: the interatomic first shell (which is often associated to the interatomic spacing typical of the liquid phase [96]) and other eventual longer-range peaks sometimes occurring as a consequence of intermolecular interactions [97]. In this situation, no sharp peak is observed, due to the disordered nature of the system. This disorder can be due to two effects:

(i)     a polydispersion of the value of the interatomic distance represented by the peak. The intrinsically-disordered nature of the system (fluid) gives a peak broadening whose width gives the distance polydispersion. The order is partially lost at any distance.

(ii)    Reduced size of the domain. The band broadening is due to the fact that the specific interatomic distance is only help at a certain length, called the correlation length. The order is lost beyond this length. The scattering domain size can be derived by the full width at half maximum (FWHM) of the band through the Debye-Sherrer formula:

$$\Delta = \frac{K\lambda}{FWHM \, \cos\theta} \tag{6}$$

where $\Delta$ is the average scattering domain size $\theta$ is the Bragg angle, $\lambda$ is the wavelength of the incident beam, the FWHM (is expressed in radians and must be corrected for instrumental broadening, and K is a factor, approximately equal to unity, related to the domain shape [98].

This approach has proved to be effective in the structural analysis of structured molecular fluids [99,100], even in ionic liquids [101], and recently also in bitumen [1]

### 6.2.3. Applications of Scattering Techniques to Bitumen

Due to the high diffusion of X-ray scattering techniques and the development of lab-scale instrumentation giving adequate data quality, X-ray scattering was used for the analysis of bitumen even in the '60 [17]. Of course, when exploring the structure, taking into account values of typical length scales longer than a few nm small-angle scattering is best suitable [102]. However, in the wide angle range, the spectrum gives already a significant amount of information: it will be now briefly shown how to interpret it taking as an example of the representative spectrum, reported in Figure 27, as a function of a scattering vector *q* ($q = (4\pi/\lambda) \sin\theta$).

Let us pay attention to the following features:

1.    A prominent broad band centered around 1.3 Å$^{-1}$ dominates the spectrum;
2.    A weak and broad band around 3 Å$^{-1}$;
3.    There is a tiny, but sharp, peak around 0.5 Å$^{-1}$, not always present;

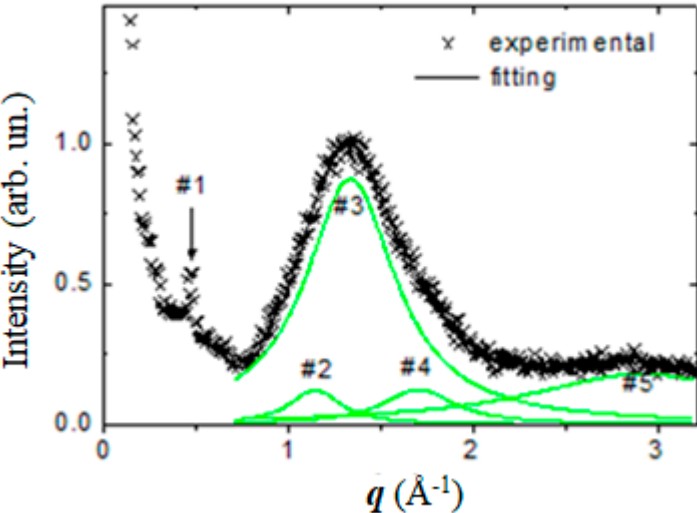

**Figure 27.** A typical X-ray scattering spectrum together with the Lorentzian deconvolution.

Different peaks (in Figure 27 indicated with progressive numbers) indicate that there are different characteristic distances belonging to various atomic and molecular organizations of different levels of complexity. The position of the center of each peak gives the characteristic distance (d), according to the Bragg law (Equation (7)) which, readapted, gives the following:

$$d = \frac{n\lambda}{2\sin\theta}.$$ (7)

As above, $d$ is the interplanar distance, $\theta$ is the scattering angle, and $\lambda$ is the wavelength of the incident radiation. Table 3 resembles the peculiarities and meaning of the peaks.

**Table 3.** Bands in the WAXS profile with the corresponding characteristics.

| Position ($\mathring{A}^{-1}$) | Features | Characteristic Distance ($\mathring{A}$) | Meaning |
|---|---|---|---|
| 1.3 | Dominant and broad | ~4.7 | combination of various intermolecular distances between alkyl and aromatic parts: See text |
| 2.9 | Broad and weak | 2.2 | interatomic distance within asphaltene; |
| ~0.5, varying | Not always present usually tiny | ~13 | supra-molecular aggregation: Repetition distance of aggregates of local asphaltene aggregates |

The most prominent band centered at around 1.3 $\mathring{A}^{-1}$ deserves some attention. In alkyl-based fluids, this band has been attributed to a characteristic intermolecular lateral distance of 4.4–4.7 $\mathring{A}$ [96,100,103] usually present in the conventional liquid (disordered) state [96,97,99–105]. Aromatic compounds are characterized by shorter distances, due to their tendency to form stacks, so they give the so-called graphene band (the lateral distance of about 3.6 $\mathring{A}$) [22]. Due to the simultaneous presence of both aliphatic and aromatic compounds in the bitumen, it is reasonable to treat this band as a not-resolved superposition of these two contributions. Deconvolution in terms of bell-shaped curves is sometimes necessary to discriminate all the signals. Two noteworthy observations are due:

1.  The fitting procedure will also help in analyzing the weaker band at higher angles (around 2.9 $\mathring{A}^{-1}$ and which is a characteristic distance $d$ of about 2.2 $\mathring{A}$) which is sometimes partially overlapped;
2.  The fitting allows to derive, from each curve, also the Full Width at Half Maximum (FWHM)

The shorter distance, of about 2.2 $\mathring{A}$, can be surely attributed to some particular interatomic distance. The value is in the range of the typical distance between non-adjacent carbons reported for polycyclic aromatic compounds [106] so such attribution can be safely hypothesized. In the range

0.3–0.8 Å$^{-1}$ it can be present at a tiny peak. This would reveal the occurrence of a supra-molecular aggregation and would be, therefore, associated to a repetition distance between one asphaltene local aggregate and its neighboring one [22], suggesting the presence of aggregates of asphaltene aggregates, in accordance with a model of a complex system with different levels of complexity. Finally, in the low-angles range of the WAXS spectrum ($q < 0.3$ Å$^{-1}$), the fractal aggregation of the supra-aggregates of asphaltene clusters can be explored. The presence of self-similar, fractal structures, in fact, can be in principle possible in bitumen: in these cases, interfacial boundary is not sharp, and a scaling law between the mass M (or particle number N) and the enclosed volume is established, which furnishing an indication of how efficiently the particles are packed [107]. For a porous fractal cluster containing N identical primary units this scaling law is expressed as $N \sim R^{D_f}$, were the fractal dimension $D_f$ is connected with the involved aggregation mechanism. Since 1984 scattering techniques have been widely used for characterization of materials having fractal microstructures. The fractal dimension of a particle can be determined by analyzing the power-law regime of the scattered intensity $I(q) \sim q^{-\alpha}$, where the exponent $\alpha$ is related to the fractal dimension $D_f$ of the scattering structures. For a mass fractal, it is possible to show that $\alpha = D_m$ and $1 < \alpha < 3$ in a three-dimensional space. In contrast, $\alpha = 6 - D_s$ for surface fractals. If $D_s = 2$ the well-known Porod's law $I(q) \sim q^{-4}$ for non-fractal structures with smooth interfaces is obtained.

So, the slope $\alpha$ can be, therefore, derived by linear fit, in an adequate region of the spectrum (see Figure 28 as reference), using the equation:

$$I(q) \propto q^{-\alpha} \tag{8}$$

However, self-similar hierarchical structures are seldom present or rarely probed in the literature, and, due to diversity of bitumen origin, chemical composition and age, this feature of the structure can change dramatically. Additives can change (i) the fractal structure, if present (ii) the band around 1.3 Å$^{-1}$ and (iii) the peak around 0.5 Å$^{-1}$, if present, but not the higher angle band around 2.2 Å$^{-1}$ because this distance refers to an intra-molecular distance, and it is not expected, therefore, to be changed by the presence of an additive.

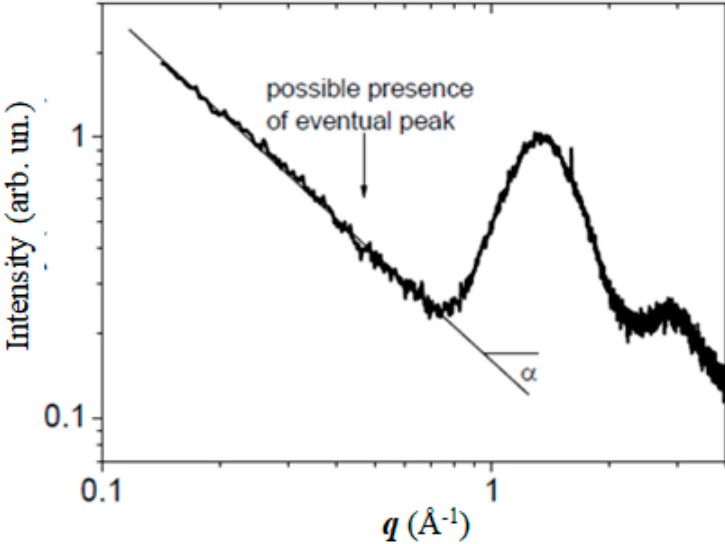

**Figure 28.** X-ray scattering spectrum of a bitumen showing fractal-like structure (see text for details).

### 6.2.4. Relaxometry Nuclear Magnetic Resonance Theory

The second vanguard technology that can be used to analyze the bitumen is the Relaxometry NMR. This method probes dynamic features and consists of measuring the $^1$H-NMR relaxation times of bitumen at the low magnetic field. In the current high-resolution NMR technique, it is not

possible to obtain high resolution spectra because of their highly heterogeneous and low magnetic field strengths. In a basic NMR concept, at equilibrium, protons nuclei are distributed among the energy levels according to a Boltzmann distribution. Following any process that disrupts this distribution (e.g., absorption of radio frequency energy), the nuclear spin system returns to equilibrium with its surroundings (the "lattice") by a first-order relaxation process characterized by a time $T_1$ called the spin−lattice relaxation time. To account for processes that cause the nuclear spins to come to equilibrium with each other, a second time $T_2$ is required. $T_2$ is called the spin−spin relaxation time, because the relaxation is concerned with the exchange of energy between spins via a flip-flop type mechanism. In a perfectly homogeneous field, the NMR time constant of the decay would be $T_2$, but, in fact, the signal decays in a time $T_2$* that often is determined primarily by field inhomogeneity, since nuclei in different parts of the field precess at slightly different frequencies, and hence, quickly get out of phase with each other. Thus, the signal decays with a characteristic time $T_2$*. This decay directly measures the decrease in the transverse magnetization $M_{xy}$. The contribution of the magnetic field inhomogeneity to the free induction decay precludes the use of this decay time, $T_2$* as a measure of $T_2$. A method for overcoming the inhomogeneity problem is to apply the Carr−Purcell technique (CP) [108]. This method may be described as a 90°, $\tau$, 180°, 2$\tau$, 180°, 2$\tau$, 180°, 2$\tau$, ... pulse sequence. In Figure 29, experimental steps of bitumen, NMR Relaxometry are shown. Four hundred echoes have been obtained by Carr−Purcell sequence and analyzed by Inverse Laplace Transform (ILT).

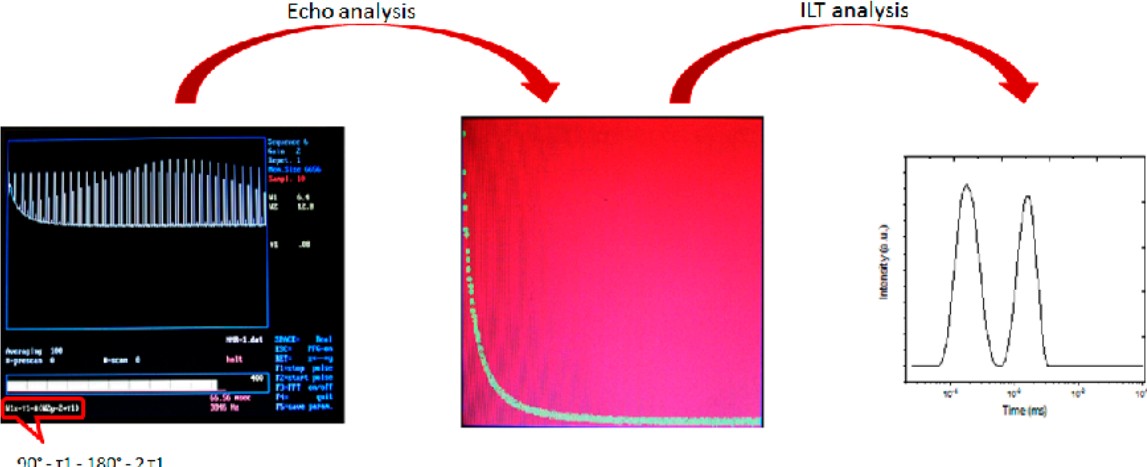

**Figure 29.** Schematic sequence of Echo and Inverse Laplace Transform (ILT) analysis in NMR Relaxometry.

The 180° pulses are applied at 90° phase difference relative to the initial 90° pulse, and the $\tau$ time delay was 0.05 ms. The main advantage of such a multiple echo technique is its quickness compared with other techniques based on single echoes. Consequently, it allows multiple accumulations of the echo train signal, which is an important issue in low-field experiments where the detection sensitivity is strongly reduced relative to the high-field experiments. If the CP envelope has a mono-exponential decay, the relaxation time $T_2$ of the sample can be obtained by fitting the *n* data to the following equation:

$$A_n = A_0 e^{-\frac{2\pi\tau}{T_2}} \tag{9}$$

where $A_n$ is the amplitude of the *n*th echo in the echo train and $A_0$ is a constant depending on the sample magnetization, filling factor, and other experimental parameters. Usually, the $T_2$ relaxation time varies all over the sample because of the sample heterogeneity or surface relaxation differences; then a multiexponential attenuation of the CP envelope should be observed. Hence, if inside the sample, a continuous distribution of relaxation times exists, the amplitude of the *n*th echo in the echo train is given by:

$$A_n = A_0 \int_0^\infty P(T_2)e^{-\frac{2\pi\tau}{T_2}} \, dT_2, \tag{10}$$

where P($T_2$) is the $T_2$ relaxation time probability density.

Equation (10) suggests that the analysis of the experimental data using an inverse Laplace transform (ILT) might provide the relaxation time probability function. The ILT is a well-known mathematical tool, where it needs to face the inverse problem of estimating the desired function from the noisy measurements of experimental data. For convenience, the definition of the problem will be shortly recalled. Let $f(t)$ be a function defined for $t \geq 0$; the function $F(s)$ introduced by means of the expression:

$$F(s) = L\{f(t)\} = \int_0^\infty f(t)e^{-st} \, dt \tag{11}$$

is the real Laplace transform of $f(t)$. The inverse process, indicated by the notation $f(t) = L^{-1}[F(s)]$, is termed the inverse Laplace transform (ILT). $P(T_2)$ is the ILT of the unknown function that fit the echo amplitude curve. Hence, $P(T_2)$ can be understood as a distribution of rate (inverse of time) constants, strictly speaking, a Probability Density Function (PDF) that, among other things, could account for the different macro-structures that compose the bitumen binder [109]. This technique allows finding the PDF distribution, which associates with relaxation times that correspond to unrelated molecular aggregates inside the bitumen.

### 6.2.5. Applications of Relaxometry Nuclear Magnetic Resonance to Bitumen

Generally, regarding types of bitumen, the $T_2$ relaxation time distribution exhibits two peaks. Direct correlation can be made between $T_2$ and the rigidity of structures in these materials [43], as well as the molecular constraint, which causes dynamic hindrance [46]. The shorter $T_2$ times (around 10 ms) reasonably corresponds, therefore, to more rigid supra-molecular aggregates; hence, they are attributed to asphaltenes. Conversely, high $T_2$ times (around 100 ms) can be attributed to low intra-molecular interactions; they can be referred to the maltene fraction of the sample under examination. This finding supports the colloidal model of the bitumen. In fact, if the polar fluid model suggested by Christensen [110] were applicable to our system, the ILT result would demonstrate a sole broad peak referred to as a continuous $T_2$ time. Figure 30 shows a typical Probability Density Function obtained through an ILT transform of $T_2$ relaxation time data determined by analyzing neat and modified 100/130 penetration grade bitumen supplied by Highway Research Institute (Almaty, Kazakhstan) [111]. The measures of $T_2$ were made at a temperature 15 °C lower than transition temperature (solid-liquid) measured by dynamic temperature ramp test experiment. In Figure 30 the panels refer to the following samples: SD refers to PAV bitumen + 2 wt% HR (the green rejuvenator), SC refers to PAV bitumen + 2 wt% VO (Vegetable Flux Oil), SB PAV bitumen, SA Neat Bitumen.

The $T_2$ relaxation time distribution can be considered a structural fingerprint of the bitumen where changes in the $T_2$ relaxation times evidence modification in the structure of colloidal binder. In particular, the powerful ILT, applied to the echo decay, can be used to verify the effectiveness of the real rejuvenator. The same or better similar profile of $T_2$ relaxation times prove that the structure of aged bitumen has been regenerated reorganizing the same distribution of asphaltene micelles in the maltene phase. In general, by ILT NMR relaxometry it is possible to follow the structural evolution of bitumen when additives (polymer surfactants, etc.) are added, in fact, the relaxation distribution is strongly affected by the supramolecular organizations present in the colloids.

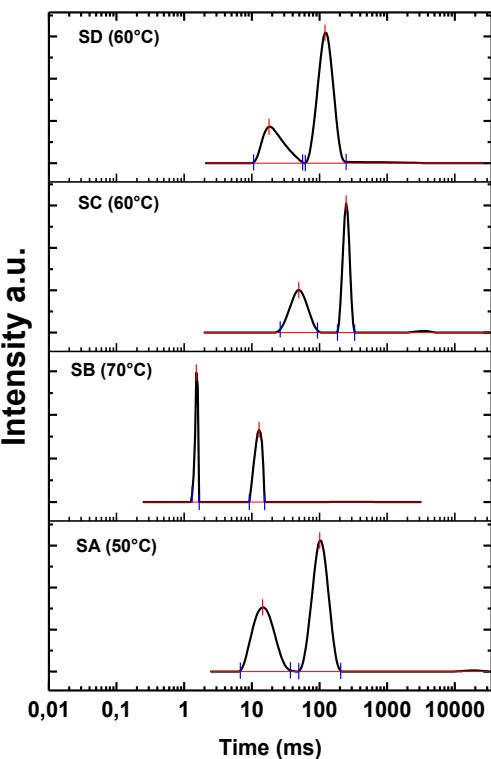

**Figure 30.** Inverse Laplace Transform (ILT) showing the Probability Density Function (PDF) of asphaltene and maltene aggregates of bitumen samples.

## 7. Concluding Remarks

1. Many materials are used for rejuvenating bitumen by lowering viscosity and stiffness. Since different physico-chemical mechanisms are involved in bitumen ageing (oxidation, evaporation, structural changes), different mechanisms of actions can be consequently exerted by the various rejuvenators (softening/fluxing, restoration of the pristine structure/properties). The distinction of the various mechanisms has been highlighted. These aspects have been shown in Sections 1–3.

2. The state-of-the-art constituted by the works carried out by several researchers in this field has been shown. Low-cost oils are generally added to increase the maltene fraction, but it should be noticed that an additive having complete rejuvenating function should also induce a reorganization of the chemical structure of asphaltenes and their assemblies. The restoring of the aged bitumen structure to the original conditions is not trivial, due to its complex organization at the supra-molecular scale. This has been extensively shown in Section 4.

3. Taking into account the complex chemistry involved in the bitumen rejuvenation, the additive performances can be improved by chemical manipulation/modification (paragraph 5.1). Some perspectives have been also presented (Section 5.2) considering for the complexity of the systems and suggesting the use of amphiphilic species as promising rejuvenator thanks to their simultaneous presence, within their molecular architecture of both polar and apolar moieties which permits their simultaneous interactions with polar (asphaltene clusters), apolar (maltene) and amphiphilic (resins) species of the bitumen.

4. Scattering techniques and nuclear magnetic relaxometry have been presented as vanguard and promising techniques deserving attention for deeper analyses in bitumen. In fact, they can probe the effectiveness of a rejuvenator in restoring the microstructure of bitumen after the aging process, whereas, mechanical properties, on their own, are not enough for investigating this aspect. A clear introduction to the physics of the techniques and applications to the study of bitumen has been presented.

5. With this work, we would like to share with the reader our belief that the detailed analysis of the physics of bitumen at the molecular basis extends the information taken from the commonly used empirical and quick tools. This allows to better understand the phenomena taking place in bitumen furnishing new tools for the piloted design of new and ever-performing rejuvenators.

6. We wanted to furnish a novel viewpoint for the study of bitumen based on the concepts of the complex systems in physics. According to this approach, the final behavior of the material is not only dictated by specific interactions, as usually assumed in most of the research papers, but also by collective contributions of many molecules interacting and aggregating themselves usually at different length scales in hierarchical structures generating emerging properties. We hope that this study can constitute a novel approach for the investigation of bitumen, and the improvements of its performances.

**Author Contributions:** P.C. (Pietro Calandra) conceptualization methodology and writing, P.C. (Paolino Caputo) conceptualization; M.P. editing correction and writing, V.L. conceptualization and research, R.A. conceptualization, and C.O.R. conceptualization and supervision.

**Funding:** This research received no external funding.

**Conflicts of Interest:** The authors declare no conflict of interest.

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
