# Peer review of "A Review on Bitumen Rejuvenation: Mechanisms, Materials, Methods and Perspectives"

_applsci, doi:10.3390/app9204316_

Round 1
Reviewer 1 Report
The nicely written paper deals with defining fluxing/rejuvenating of the bitumens with special additives. Paper presents chemical analysis and physical impact of various additieves on the bitumen properties. Overal, paper is ruther complex and usefull study.
I would recommend Authors to familirize themself with some of the papers shown below, as they can provide some additional point of view in the analysis on the mechanism of rejuvenation and its impact on the asphalt mixture properties:
Majid, Z.; Esmaeil, A.; Hallizza, A.; Mohamed, R.K. Investigation of the possibility of using waste cooking oil as a rejuvenating agent for aged bitumen. J. Hazard. Mater. 2012, 233, 254–258.
Król J. B., Niczke Ł., Kowalski K. J., “Towards understanding polymerization process in bitumen bio-fluxes“, Materials, vol. 10, 2017.
Zaumanis, M.; Mallick, R.B.; Poulikakos, L.; Frank, R. Influence of six rejuvenators on the performance properties of Reclaimed Asphalt Pavement (RAP) binder and 100% recycled asphalt mixtures. Constr. Build. Mater. 2014, 71, 538–550.
Kowalski K. J., Król J. B., Bańkowski W., Radziszewski P., Sarnowski M., “Thermal and Fatigue Evaluation of Asphalt Mixtures Containing RAP Treated with a Bio-Agent”, Applied Sciences, vol. 7, no. 3: 216, 2017. Special issue: Advanced Asphalt Materials and Paving Technologies.
Bocci, E.; Grilli, A.; Bocci, M.; Gomes, V. Recycling of high percentages of reclaimed asphalt using a bio-rejuvenator—A case study. In Proceedings of the 6th Eurasphalt & Eurobitume Congress, Prague, Czech, 1–3 June 2016; Paper No. EE.2016.334.
Somé, C.; Pavoine, A.; Chailleux, E.; Andrieux, L.; DeMarco, L.; Da Silva Philippe, B.S. Rheological behavior of vegetable oil-modified asphaltite binders and mixes. In Proceedings of the 6th Eurasphalt & Eurobitume Congress, Prague, Czech, 1–3 June 2016; Paper No. EE.2016.222.
Król J. B., Kowalski K. J., Niczke Ł., Radziszewski P., ”Effect of bitumen fluxing using a bio-origin additive”, Construction and Building Materials, vol. 114, 2016, pp. 194-203.
Tanghe, T.; Lemoine, G.; Nösler, I.; Kloet, B. Influence of rejuvenating additives on recycled asphalt (RAP) properties. In Proceedings of the 5th Eurasphalt & Eurobitume Congress, Istanbul, Turkey, 13–15 June 2012; Paper No. P5EE-416.
Author Response
We thank the referee for his/her kind words.
One of the suggested references (Majid et al., J. Hazard. Mater. 2012, 233, 254–258) had been already cited in the unrevised version of our manuscript at the reference number 47.
We have read the other papers, finding some of them very interesting. We found them useful for increasing the quality of our manuscript: their use suggested even a new section within paragraph 4.
In the specific, we have added the following references to our manuscript
[Bocci E.; Grilli A.; Bocci M.; Gomes V., Recycling of high percentage of reclaimed asphalt using bio-rejuvenator- a case study, In: Proceedings of the 6th Eurasphalt & Eurobitume Congress, Prague, Czech, 1–3 June 2016; Paper No. EE.2016.334]
[Tine T.; Lemoine G.; Nösler I.; Kloet B., Influence of rejuvenating additives on recycled asphalt (RAP) properties. In: Proceedings of the 5th Eurasphalt & Eurobitume Congress, Istanbul, Turkey, 13–15 June 2012; Paper No. P5EE-416.]
[Król J. B.; Kowalski K. J.; Niczke L; Radziszewski P, Effect of bitumen fluxing using a bio-origin additive, Construction and Building Materials 2016, 114, 194-203]
[Somé C.; Pavoine A; Chailleux E; Andrieux L; DeMArco L; Philippe Da -S; Stephan B; Rheological behaviour of vegetable oil-modified asphaltite binders and mixes, In: Proceedings of the 6th Eurasphalt & Eurobitume Congress, Prague, Czech, 1–3 June 2016; Paper No. EE.2016.222]
Reviewer 2 Report
Comments and Suggestions for Authors:
In this paper the review concerned the bitumen rejuvenation problem was discussed. The reviewer would like to thank the Authors for their efforts. The general feeling is that the paper seems to be too long. The total length of the paper (47 pages, 1458 lines) made sometime hard to understand the general idea of the paper and intentions of Authors. Many interesting findings but scattered in sometimes random places of the paper make it impossible to summarize. In reviewer opinion Authors also had the problem how to do that because the Paragraph 6 “Concluding remarks” is very general and has only 15 lines!
The detailed comments are as follows:
The title of the paper is not correct. Firstly it should be “bitumens” not “bitumes”. Secondly the title doesn’t sound correctly according to general idea of the paper. Please consider to change the title. Maybe better is: “Bitumen rejuvenation. A review on materials and methods”? Please try to avoid the style: “we have tried”, “we have made” and so on. Better is: “it was made”. In many places in the paper the Authors used the term: “asphalt” or “bitumens and asphalts” probably describing the same material. The reviewer have checked the previous papers of the Authors, for example paper published in February in Applied Sciences”, titled: “Bitumen and Modification: A Review on Latest Advances”. In that paper was used the term “bitumen” that is more correct according to European terminology defined the material produced from the crude oil. In line 28 and also 61 the Authors described the asphalt concretes as a general description of asphalt mixtures. There is incorrect. Asphalt concrete is only one of the many types of asphalt mixtures that are use in road construction. In reviewer opinion it should be term “asphalt mixture” or even asphalt concrete but please to be precise and define that mixture. The Introduction paragraph doesn’t fulfil all requirements according to that point. The problem was not explained enough. Even if the idea of the paper is as the review paper the Introduction should be more detailed. There is very large disproportion between the Introduction and the other paragraphs. The objectives of the paper are very general and were not clear defined. In the paragraph 2 “Materials and Methods” was very hard to understand what kind of materials and methods were used in the research. Please try to organize this paragraph firstly explaining the materials and secondly discussing the methods. Many Figures were reprinted from other papers. For reviewer it was hard to check and detect plagiarism in the text. But that issue is very important.
Author Response
REFEREE#2
Here is the point-by-point list of the corrections made according to the referee’s criticisms:
The title of the paper is not correct. Firstly it should be “bitumens” not “bitumes”. Secondly the title doesn’t sound correctly according to general idea of the paper. Please consider to change the title. Maybe better is: “Bitumen rejuvenation. A review on materials and methods”?Reply: “Bitumes” in the title was just a typo and we apologize for this. No other “bitume” or “bitumes” is present in the manuscript. We like the referee’s suggestion so we have changed the title, accordingly, to “A review on bitumen rejuvenation: mechanisms, materials, methods and perspectives”. Please note that we have added “perspectives” in the title to emphasize our critical work in commenting the literature works and in suggesting new methods of analysis as well as the new approach of treating bitumens within the conceptual framework of physics of complex systems.
Please try to avoid the style: “we have tried”, “we have made” and so on. Better is: “it was made”.
Reply: We have checked the manuscript throughout. We changed such expressions according to the referee suggestion.
In many places in the paper the Authors used the term: “asphalt” or “bitumens and asphalts” probably describing the same material. The reviewer have checked the previous papers of the Authors, for example paper published in February in Applied Sciences”, titled: “Bitumen and Modification: A Review on Latest Advances”. In that paper was used the term “bitumen” that is more correct according to European terminology defined the material produced from the crude oil.
Reply: Bitumens and asphalts are certainly different things, as highlighted in par. 1 Introduction and we are sorry for having generated this confusion. In the revised version we have paid careful attention in making clear distinction between the two. As it can be seen (see highlighted changes in the revised manuscripts) the terms “asphalts” and “bitumens” have been used in a proper way.
The difference has been also stressed when showing more explicitly the scope of the work: in the introduction paragraph we added the sentence (line 77-79): “This works will show the state-of-the-art in the use of rejuvenators in bitumens, taking care to highlight also some applications to the rejuvenation of recycled asphalts for a more complete view of the problematics”.
Moreover, we have added (end of paragraph 4.1 – general requirements) the sentence “This paragraph, from the next sub-paragraph on, is devoted to show the works involving rejuvenators as additives in bitumens. Some applications to the rejuvenation of partly recycled asphalts will be also shown for a more complete view of the problematics also because for practical uses sometimes rejuvenation must be exerted on recycled asphalts (RAP, reclaimed asphalt pavement).”
In line 28 and also 61 the Authors described the asphalt concretes as a general description of asphalt mixtures. There is incorrect. Asphalt concrete is only one of the many types of asphalt mixtures that are use in road construction. In reviewer opinion it should be term “asphalt mixture” or even asphalt concrete but please to be precise and define that mixture.
Reply: We thank the referee for the clarification. As expressed in the previous reply to referee’s comment, we have corrected the manuscript accordingly, ruling out possibilities of confusion. The main topic of the review is bitumens rejuvenation, so the overall work maintains, in the description of the physico-chemical processes of ageing and in those involved in rejuvenation, a general description of the bitumen to focus attention on the scientific point of view which is of general validity, giving to asphalts a secondary importance. However, also reclaimed asphalts have been discussed in the “state-of-the-art” paragraph as examples of rejuvenated systems for a more complete view, but this difference has been stressed in several points (see reply to previous comment)
5a. The Introduction paragraph doesn’t fulfil all requirements according to that point. The problem was not explained enough. Even if the idea of the paper is as the review paper the Introduction should be more detailed.
Reply: We have distributed all the introductory information in the first three paragraphs, in order to avoid overloading of information in one introductory paragraph only. So, the “introduction” (paragraph 1) is a very preliminary paragraph, necessary to present the problematics: it introduces to bitumens and to the phenomenon of ageing. Then the methods for bitumen characterization and the ageing/rejuvenation processes have been discussed in the successive two paragraphs. These three paragraphs, therefore, are to be intended as the introductory part prior to the “state-of-the-art” which is, now, also subdivided for better readability. Note that we changed also the names of the first paragraphs:
Introduction bitumens and ageing Methods for bitumens characterizations mechanisms of ageing and rejuvenatingfor a more explicit presentation of the topics.
It must also be noted that the clear distinction of the two mechanisms of rejuvenation, generally not-well-defined in the literature, in a novelty of our work and constitutes, in our opinion, its added value, allowing also us to propose novel experimental techniques for their discrimination (paragraphs 6). Following the suggestion of the referee, in the revised version the introduction section better defines the structure and the aim of the entire review, also with the indication of the pertinent paragraphs for a better readability (end of pargraph 1 – lines 78-85). Also, the abstract has been rewritten accordingly as well as the final paragraph of “concluding remarks”.
5b. There is very large disproportion between the Introduction and the other paragraphs.
Reply: We are not sure about the referee’s opinion, and in particular if the introduction seemed too long or too short (“disproportion” in what sense?). In any case, the first three paragraphs, which give all the necessary information to better follow the state-of-the-art section, have been renamed and in part rewritten. Probably agreeing with the referee, also, we were conscious that the paragraph dealing with the state-of-the-art was too long and maybe needing division in sub-paragraphs for better readability. In the revised version, also to take into account for the works suggested by referee #1, we have divided this part into sub-paragraphs for a better presentation of the logical sequence of our discussion. We hope that in the present form all the parts can look more harmonic.
The objectives of the paper are very general and were not clear defined.
Reply: We thank the referee for his/her criticism. In the revised version, as stated in the reply to referee’s comment 3 (where some new sentences of the revised manuscript have been reported), we have better pointed out the objectives of the paper in the abstract (which has been entirely rewritten), at the end of the introduction (lines 77-85), at the end of pargraph 4.1, and in the concluding remark section (see highlighted parts). It must be pointed out that, differently from common review papers, our manuscript, besides presenting the state-of-the-art (paragraph 4), it presents a clear, new and marked distinction between different “rejuvenators” according to their effect and influence at the microscopic length-scale (paragraph 3 “mechanisms of ageing and rejuvenating” and sub-paragraphs) individuating techniques and methods of analysis for their distinction and, also, showing methods for the synthesis of ever more performing rejuvenators (paragraph 5) and suggesting vanguard techniques of investigation (paragraph 6). This is, in our opinion a quite added-value aspect of our work deserving attention. This additional aim of the work, has also been highlighted in the revised version of the manuscript (end of introduction, see highlighted part).
Furthermore, in the revised version of the manuscript we have taken care in emphasizing an aspect probably not underlined enough in the unrevised version: as stated in the abstract “… the work represents a critical analysis of the state-of-the-art taking into account for the molecular basis at the origin of the observed behavior: furnishing a novel viewpoint for the study of bitumens based on the concepts of the complex systems in physics, it constitutes a novel approach for the study of these systems.”. This concept has been reinforced also in the concluding remarks sections (items 5 and 6)
In the paragraph 2 “Materials and Methods” was very hard to understand what kind of materials and methods were used in the research. Please try to organize this paragraph firstly explaining the materials and secondly discussing the methods.
Reply: We thank the referee for his precious suggestion. We admit that a paragraph name like “materials and methods” is typical of a research paper. Our work is instead a review, so we understand to have ingenerated some confusion, and we are sorry for this. In response to the referee’s comment we have made clearer both the aspects of the “materials” object of the present review (bitumens) and the “methods” followed by the scientific community for their study: some parts of the paragraph 2 previously called “materials and methods” have been moved to the paragraph 1 since it is now better devoted to the description of the material (bitumens). Consequently, paragraphs 2 is now entitled “methods for bitumens characterizations” explicitly dealing with the “methods” used for bitumens study. The order of the two, also, matches that suggested by the referee.
Many Figures were reprinted from other papers. For reviewer it was hard to check and detect plagiarism in the text. But that issue is very important.Reply: We have taken the utmost care in asking permissions for figure reproduction. Figures 1, 7, 26, 27, 28, 29, 30 were made by ourselves; figure 21 was readapted by us from literature (see caption). For all the others we have the permission for reproduction, which is highlighted in all the relative figure captions. As for plagiarism, It must be noted that we have made the effort to summarize in few words the work made by all the Authors cited, trying to be clear and exhaustive. This, not only ruled out any risk of plagiarism which we definitely condemn, but also gave us easy opportunity to sometimes express our opinion on the works carried out by the other researchers (see for example, in line 741 our opinion on the good job done by Nahar et al.)
We thank the referee for his/her suggestions.
Round 2
Reviewer 2 Report
Thank you for all your improvements of the paper.